# Semaphorin7A patterns neural circuitry in the lateral line of the zebrafish

Agnik Dasgupta[1]*, Caleb C Reagor[1,2], Sang Peter Paik[1†], Lauren M Snow[1‡], Adrian Jacobo[1,3], AJ Hudspeth[1]

[1]Howard Hughes Medical Institute and Laboratory of Sensory Neuroscience, The Rockefeller University, New York City, United States; [2]Tri-Institutional PhD Program in Computational Biology and Medicine, New York, United States; [3]Chan Zuckerberg Biohub San Francisco, San Francisco, United States

*For correspondence:
agnik.dasgupta@tcgcrest.org

Present address: †Department of Functional Neuroanatomy, Institute for Anatomy and Cell Biology, Heidelberg University, Heidelberg, Germany; ‡Brigham and Women's Hospital, Boston, United States

Competing interest: The authors declare that no competing interests exist.

## eLife assessment

The **valuable** findings by Dasgupta et al demonstrate the role of Sema7a in fine tuning the morphology of the microcircuit between afferent axons and sensory hair cells in the lateral line organ. The loss and gain of function evidence provides **solid** support for a role for Sema7a in this process. Additional work is needed to determine the role for different isoforms in Sema7a-mediated synapse formation and chemoattraction as well as cell type specificity.

**Abstract** In a developing nervous system, axonal arbors often undergo complex rearrangements before neural circuits attain their final innervation topology. In the lateral line sensory system of the zebrafish, developing sensory axons reorganize their terminal arborization patterns to establish precise neural microcircuits around the mechanosensory hair cells. However, a quantitative understanding of the changes in the sensory arbor morphology and the regulators behind the microcircuit assembly remain enigmatic. Here, we report that Semaphorin7A (Sema7A) acts as an important mediator of these processes. Utilizing a semi-automated three-dimensional neurite tracing methodology and computational techniques, we have identified and quantitatively analyzed distinct topological features that shape the network in wild-type and Sema7A loss-of-function mutants. In contrast to those of wild-type animals, the sensory axons in Sema7A mutants display aberrant arborizations with disorganized network topology and diminished contacts to hair cells. Moreover, ectopic expression of a secreted form of Sema7A by non-hair cells induces chemotropic guidance of sensory axons. Our findings propose that Sema7A likely functions both as a juxtacrine and as a secreted cue to pattern neural circuitry during sensory organ development.

## Introduction

Pathfinding axons are directed to their appropriate synaptic targets by a variety of guidance cues. While approaching or traversing the target field, the growth cones of migrating axons encounter secreted or cell-surface-attached ligands and respond by steering toward or away from their sources (*Lykissas et al., 2007*; *Vaccaro et al., 2022*; *Nguyen-Ba-Charvet et al., 2008*; *Jeroen Pasterkamp et al., 2003*). Fine-grained control of these factors is critical for the establishment of proper neural circuitry.

The zebrafish's lateral line provides a tractable in vivo system for exploring the mechanisms that guide the assembly of neural circuits in a peripheral nervous system. In particular, it is possible to detect individual hair cells and the sensory axons that innervate them throughout the first week of development (*Nagiel et al., 2008*). The primary posterior lateral line on each side of the zebrafish's

tail consists of about seven neuromasts, which contain mechanoreceptive hair cells of opposing polarities—half sensitive to headward (rostrad) water motion and the complementary half to tailward (caudad) water motion—that cluster at their centers (*Jacobo et al., 2019*). The sensory axons of the lateral-line nerve branch, arborize, and consolidate around the basolateral surfaces of the hair cells (*Figure 1A*). To aid in an animal's swimming behavior, these hair cells sense water currents and relay signals to the brain through the sensory axons (*Valera et al., 2021*). The well-defined structure of the ramifications and synaptic contacts of the sensory axons highlight the need to determine the molecular signals behind the assembly of such precise microcircuits (*Dow et al., 2018*).

We adopted a candidate-gene strategy to seek factors that direct the growth of afferent growth cones in the zebrafish's lateral line. Analysis of single-cell RNA sequencing data identified the *semaphorin7a* (*sema7a*) gene to be highly expressed in hair cells of that organ (*Lush et al., 2019*). Semaphorins are important regulators of axonal growth and target finding during the patterning of diverse neural circuits (*Carulli et al., 2021*). The semaphorin family includes proteins that are secreted, transmembrane, and cell-surface-attached (*Yazdani and Terman, 2006*). Among these, Semaphorin7A (Sema7A) is the only molecule that is anchored to the outer leaflet of the lipid bilayer of the cell membrane by a glycosylphosphatidylinositol (GPI) linker (*Yazdani and Terman, 2006*). Unlike many other semaphorins, which act as repulsive cues (*de Winter et al., 2016*; *Coate et al., 2015*), Sema7A is involved in promoting axon growth (*Jeroen Pasterkamp et al., 2003*) and imparting directional signals by interacting with the integrin and plexin families of receptors residing on the pathfinding axons (*Carcea et al., 2014*; *Inoue et al., 2018*; *Inoue et al., 2021*). Sema7A likely occurs in vivo in both GPI-anchored and soluble forms (*Morote-Garcia et al., 2012*; *Nishide and Kumanogoh, 2018*; *Köhler et al., 2020*), and studies of neuronal explants confirm that both forms can induce directed axonal outgrowth (*Jeroen Pasterkamp et al., 2003*). Furthermore, it has been proposed that the GPI anchor is cleaved by membrane-resident GPI-specific phospholipases (GPI-PLs) or matrix metalloproteases to release the Sema7A into the extracellular environment and thus regulate dynamic cellular processes (*Pham et al., 2017*; *Xie and Wang, 2017*). Because these observations indicate that Sema7A might influence neuronal development both as a juxtacrine and as a diffusive signal, we investigated the role of Sema7A in sculpting a vertebrate peripheral sensory organ.

## Results

### Sema7A expression and localization in hair cells

The zebrafish genome contains a single *sema7a* gene that produces two transcripts. One transcript encodes Sema7A-GPI, a full-length, GPI-linked, cell-surface-attached protein. The second transcript yields Sema7A$^{sec}$, a protein with a truncated C-terminus, which we conjecture is secreted into the local environment. Each transcript encodes an N-terminal signal sequence and a single copy of the conserved Sema domain (*Yazdani and Terman, 2006*). To detect the presence of both *sema7a* transcripts in the hair cells of the lateral-line neuromasts, we used fluorescence-activated cell sorting (FACS) to capture the labeled hair cells (*Baek et al., 2022*) and isolated total RNA. Using primers that flank the regions encoding the conserved Sema domain and the distinct C-termini of the two transcripts, we performed reverse transcription and polymerase chain reactions (RT-PCR) to identify both the membrane-anchored and the secreted transcript in developing larvae (*Figure 1B–D*; *Figure 1—figure supplement 1A–D*). Single-cell RNA sequencing data also have shown that the GPI-linked *sema7a* transcript is particularly enriched in the immature and mature hair cells of the neuromast during early larval development (*Lush et al., 2019*), a period when sensory axons of the posterior lateral line contact hair cells and form synapses with them (*Nagiel et al., 2008*; *Dow et al., 2015*). Indeed, we have observed Sema7A to be specifically localized in both mature and immature hair cells (*Figure 1E*; *Figure 1—figure supplement 2A–D*).

As neuromasts mature, the contacts between sensory axons and hair cells increase in number, become stabilized, and establish synapses (*Nagiel et al., 2008*; *Dow et al., 2015*). Because Sema7A can modulate axon guidance (*Jeroen Pasterkamp et al., 2003*; *Carcea et al., 2014*) and synapse formation (*Inoue et al., 2018*), we wondered whether the level of Sema7A also changes during neuromast development. Upon quantifying the average intensity of Sema7A, we found similar expression in rostrally and caudally polarized hair cells of neuromasts at 1.5, 2, 3, and 4 days post fertilization (dpf). The average Sema7A intensity increased significantly over this period in both rostrally and caudally

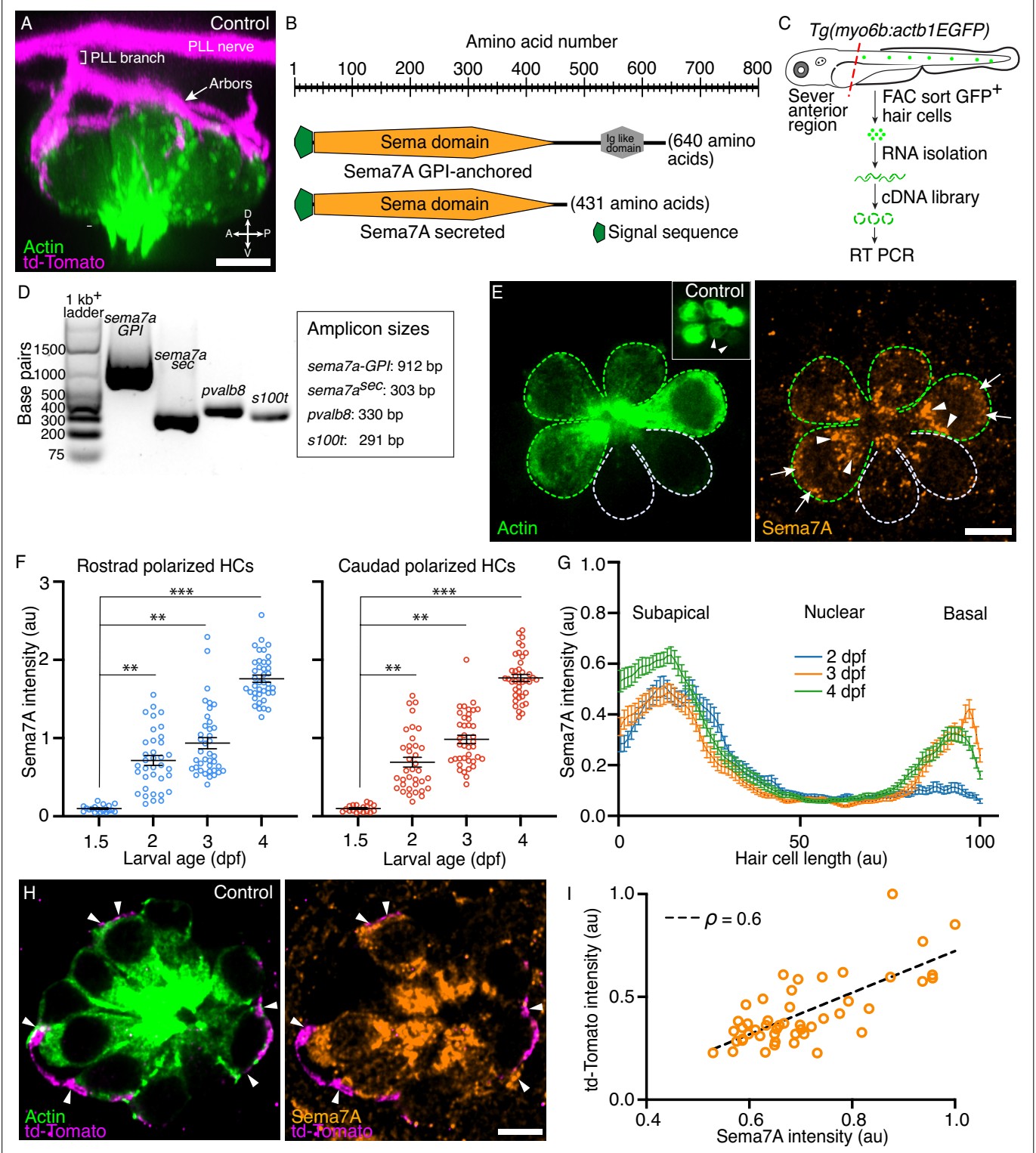

**Figure 1.** Expression of Semaphorin7A in the zebrafish's lateral line. (**A**) A volumetric rendering of a PLL neuromast depicts the sensory axons (magenta) that branch from the lateral line nerve to arborize around the basolateral surface of the hair cell cluster (green). Additional cell types in the neuromast are not labeled. Larval age, 3 dpf. A=anterior, P=posterior, D=dorsal, V=ventral. (**B**) A schematic drawing of the two variants of the Sema7A protein molecule depicts the full-length GPI-anchored form and the smaller, potentially secreted form. Both the molecules include a signal sequence (green) and a conserved sema domain (orange). (**C**) *Tg(myo6b:actb1-EGFP)* larvae at 4 dpf were terminally anesthetized and their heads were removed with fine blades. The tails were then dissociated into the component cells, which were sorted by flow cytometry. Green cells represent GFP+ hair cells. A

*Figure 1 continued on next page*

*Figure 1 continued*

cDNA library was prepared from the isolated RNA. (**D**) Gel-based RT-PCR analysis indicates the presence of both the *sema7a* transcript variants and the expression of hair cell-specific genes, parvalbumin 8 (*pvalb8*) and S100 calcium binding protein T (*s100t*). (**E**) A surface micrograph of a neuromast at 3 dpf depicts two pairs of mature hair cells (green dashed lines) and a pair of immature hair cells (grey dashed lines). Inset: among the three pairs of hair cell apices, the immature pair is indicated by arrowheads. Immunolabeling reveals that the Sema7A protein (orange) occurs consistently at the subapical region (arrowheads) and at the basolateral surface (arrows) of a hair cell. In this and in each of the subsequent neuromast images, anterior is to the left and dorsal to the top. (**F**) A plot quantifies developmental changes in the average Sema7A intensity in both rostrally and caudally polarized hair cells of neuromasts from 1.5 dpf to 4 dpf. The data stem from 18, 36, 39, and 40 hair cells in neuromasts of respectively 1.5 dpf, 2 dpf, 3 dpf, and 4 dpf larvae. (**G**) A plot quantifies the distribution of average Sema7A intensity along the hair cell's apicobasal axis. The results stem from 52, 57, and 54 hair cells of neuromasts from 2 dpf, 3 dpf, and 4 dpf larvae. (**H**) An immunofluorescence image at the nuclear level of a 4 dpf neuromast shows the contact of the sensory arbors (magenta) with the basolateral surface of the hair cells (green). Immunolabeling for Sema7A (orange) reveals enrichment of the protein at the hair cell bases and sensory-axon interfaces (arrowheads). (**I**) At the base of the hair cell, association of the td-Tomato$^+$ sensory arbors positively correlated with the Sema7A intensity. The results stemmed from 48 hair cells of 13 neuromasts from 3 dpf larvae. HC, hair cell; scale bars, 5 μm; au, arbitrary unit; means ± SEMs; $\rho$, Spearman's correlation coefficient; *** implies p<0.001 and ** implies p<0.01.

The online version of this article includes the following source data and figure supplement(s) for figure 1:

**Source data 1.** The source data contains raw uncropped gel image of the RT PCR experiment.

**Source data 2.** Sema7A intensity distribution at diverse stages of hair cell development.

**Figure supplement 1.** Isolation of GFP-expressing hair cells.

**Figure supplement 2.** Expression of Semaphorin7A in the lateral line neuromast.

**Figure supplement 3.** Sema7A progressively enriches at the hair cell base with the sensory axons.

**Figure supplement 4.** Alignment of the mouse and the zebrafish Sema7A protein sequences generated by Clustal Omega.

polarized hair cells. At 1.5 dpf, the neuromasts harbor primarily immature hair cells that make minimal contacts with sensory axons and do not form stable association with axonal terminals (*Dow et al., 2015*). At this stage the average Sema7A intensity in hair cells remained low. By 2–4 dpf the hair cells mature to form stable contacts and well-defined synapses with the sensory axons (*Nagiel et al., 2008*; *Pujol-Martí et al., 2014*). The average Sema7A intensity in hair cells at each of these stages rose to levels that significantly exceeded those of 1.5 dpf neuromasts (*Figure 1F*). This rise in the amount of Sema7A during neuromast maturation supports a role for Sema7A in the guidance of sensory axons and their interaction with hair cells.

We additionally demonstrated an anisotropic distribution of Sema7A along the apicobasal axis of each hair cell. As neuromasts develop through 2, 3, and 4 dpf, Sema7A remained highly enriched in the subapical region of the hair cell, the site of the Golgi network (*Figure 1E*; *Figure 1—figure supplement 2E*). This localization suggests that Sema7A—like other proteins of the semaphorin family (*De Wit et al., 2005*)—is directed to the plasmalemma through Golgi-mediated vesicular trafficking. Using the transgenic line *Tg(cldnb:lyn-mScarlet)* that expresses membrane-tethered mScarlet in all cells of the neuromast (*Dalle Nogare et al., 2020*), we detected Sema7A at the plasma membrane and potentially in vesicles (*Figure 1—figure supplement 2F–I*). While Sema7A maintained a high level in the subapical region, the protein increasingly accumulated at the base of the hair cell: the Sema7A level at the hair cell base was low at 2 dpf, but increased sharply by 3 dpf and 4 dpf (*Figure 1G*; *Figure 1—figure supplement 2J*). During this period the hair cells increase in number and mature,

whereas the associated sensory axons arborize to contact multiple hair cells and form robust synaptic boutons at their bases (*Nagiel et al., 2008*; *Pujol-Martí et al., 2014*). Does progressive basal accumulation of Sema7A facilitate association of sensory axon terminals to the hair cell? To address this question, we utilized doubly labeled transgenic fish *Tg(myo6b:actb1-EGFP;neurod1:tdTomato)* that marked the hair cells and the sensory axons of the posterior lateral line with distinct fluorophores (*Figure 1A*; *Ji et al., 2018*). We quantified the average Sema7A intensity and the accumulation of sensory arbors, measured by td-Tomato intensity, at the bases of

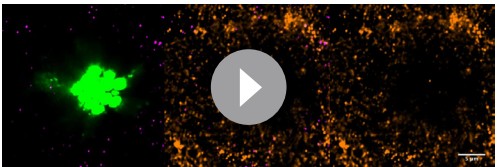

**Video 1.** Sema7A is enriched at the hair cell base and sensory-axon interface. A through-focus scan of a 4 dpf control neuromast depicts the hair cells (green) that are innervated by the sensory axons (magenta) at their basolateral surfaces where the Sema7A protein (orange) is highly enriched. Scale bar, 5 μm.
https://elifesciences.org/articles/89926/figures#video1

the hair cells. Across developmental stages, the basal association of the sensory arbors showed a positive correlation with Sema7A accumulation (*Figure 1H, I*; *Figure 1—figure supplement 3A–E*, *Video 1*). The progressive enrichment of Sema7A at the hair cell base with the sensory axons suggests that Sema7A acts as a mediator of contacts between sensory axons and hair cells.

## Sema7A restricts sensory axons around clustered hair cells

If Sema7A guides and restricts the sensory arbors at and around the hair cells clustered in a neuromast, then inactivating Sema7A function should disrupt this process. To test this hypothesis, we obtained a *sema7a* mutant allele (*sema7a^sa24691^*) with a point mutation that introduces a premature stop codon in the conserved sema domain (*Figure 2—figure supplement 1A*). The homozygous mutant, hereafter designated *sema7a^−/−^*, should lack both the secreted and the membrane-attached forms of Sema7A. The average intensity of Sema7A immunolabeling in hair cells diminished by 61% in *sema7a^−/−^* larvae (0.34±0.01 arbitrary units) in comparison to controls (0.87±0.01 arbitrary units). In a few cases, we observed minute accumulations of Sema7A at the subapical region, but little or no protein at the base of the mutant hair cells (*Figure 2A–E*). We quantified the number of hair cells in the neuromasts of the control and the *sema7a^−/−^* larvae. We did not observe significant changes in the hair cell number between the genotypes across developmental stages (*Figure 2—figure supplement 1B*). Furthermore, we tested the mechanotransduction ability of *sema7a^−/−^* mutant hair cells by labeling with the styryl dye FM 4–64 (*Nagiel et al., 2008*). The hair cells of the 4 dpf *sema7a^−/−^* larvae incorporated the FM 4–64 as effectively as the age-matched control animals. We quantified the FM 4–64 intensities from 60 hair cells and six neuromasts from three control larvae, and from 60 hair cells and six neuromasts from three *sema7a^−/−^* mutant larvae. The average intensities of FM 4–64 labeling between the genotypes did not show a significant difference, which suggests that the mutation does not impact essential hair cell functions (*Figure 2—figure supplement 1C and D*; *Videos 2 and 3*).

To characterize the impact of the *sema7a^−/−^* mutation on arbor patterning, we utilized doubly labeled transgenic fish *Tg(myo6b:actb1-EGFP;neurod1:tdTomato)* that distinctly labeled the hair cells and the sensory arbors (*Ji et al., 2018*). In control neuromasts, the sensory axons approached, arborized, and consolidated around the bases of clustered hair cells (*Figure 2F*). Although the sensory axons approached the hair cells of *sema7a^-/-^* neuromasts, they displayed aberrant arborization patterns with wayward projections that extended transversely to the organ (*Figure 2G*). To quantitatively analyze the arborization morphology, we traced the sensory arbors in three dimensions to generate skeletonized network traces that depict–as pseudocolored trajectories–the increase in arbor length from the point of arborization (*Figure 2H, I*).

To visualize the hair cell clusters and associated arborization networks from multiple neuromasts, we aligned images of hair cell clusters, registered them at their hair bundles, from neuromasts at 2 dpf, 3 dpf, and 4 dpf (*Figure 2—figure supplement 1E and F*). Throughout development, the arborization networks of control neuromasts largely remained within the contours of the hair cell clusters; the few that reached farther nonetheless lingered nearby (*Figure 2J*; *Figure 2—figure supplement 1G1*). This result suggests that an attractive cue retained the axons near the sensory organ. In *sema7a^-/-^* neuromasts, however, the neuronal arbors failed to consolidate within the boundaries of hair cell clusters and extended far beyond them (*Figure 2K*; *Figure 2—figure supplement 1H and J*; *Video 4*).

We quantified the distribution of the sensory arbors around the center of the combined hair cell clusters. For both the control and the *sema7a^-/-^* neuromasts, the arbor densities peaked proximal to the boundaries of the hair cell clusters. Beyond the cluster boundaries, in control neuromasts at 2 dpf, 3 dpf, and 4 dpf the arbor densities fell sharply at respectively 31.7 μm, 36.6 μm, and 32.7 μm from the center. In *sema7a^-/-^* neuromasts of the same ages, the sensory arbors extended respectively 38.5 μm, 51.4 μm, and 56.6 μm from the center (*Figure 2L*; *Figure 2—figure supplement 1K and L*; *Figure 2—figure supplement 2A*). Furthermore, we quantified the degree of contact of the sensory arbors to their hair cell clusters in individual neuromasts from both control and *sema7a^-/-^* mutants. In control neuromasts, the degree of contact was 83 ± 1% (mean ± SEM) at 2 dpf and 84 ± 1% at 3 dpf, but significantly increased to 90 ± 1% at 4 dpf (*Figure 2M*; *Figure 2—figure supplement 2B*). This observation indicates that the sensory arbors reinforce their association to the hair cell clusters as neuromasts develop. However, in *sema7a^-/-^* neuromasts such contact was significantly reduced. Only 69 ± 4% of sensory arbors at 2 dpf were closely associated with the corresponding hair cell clusters, a value that remained at 67 ± 2% at both 3 dpf and 4 dpf (*Figure 2M*). These findings signify that,

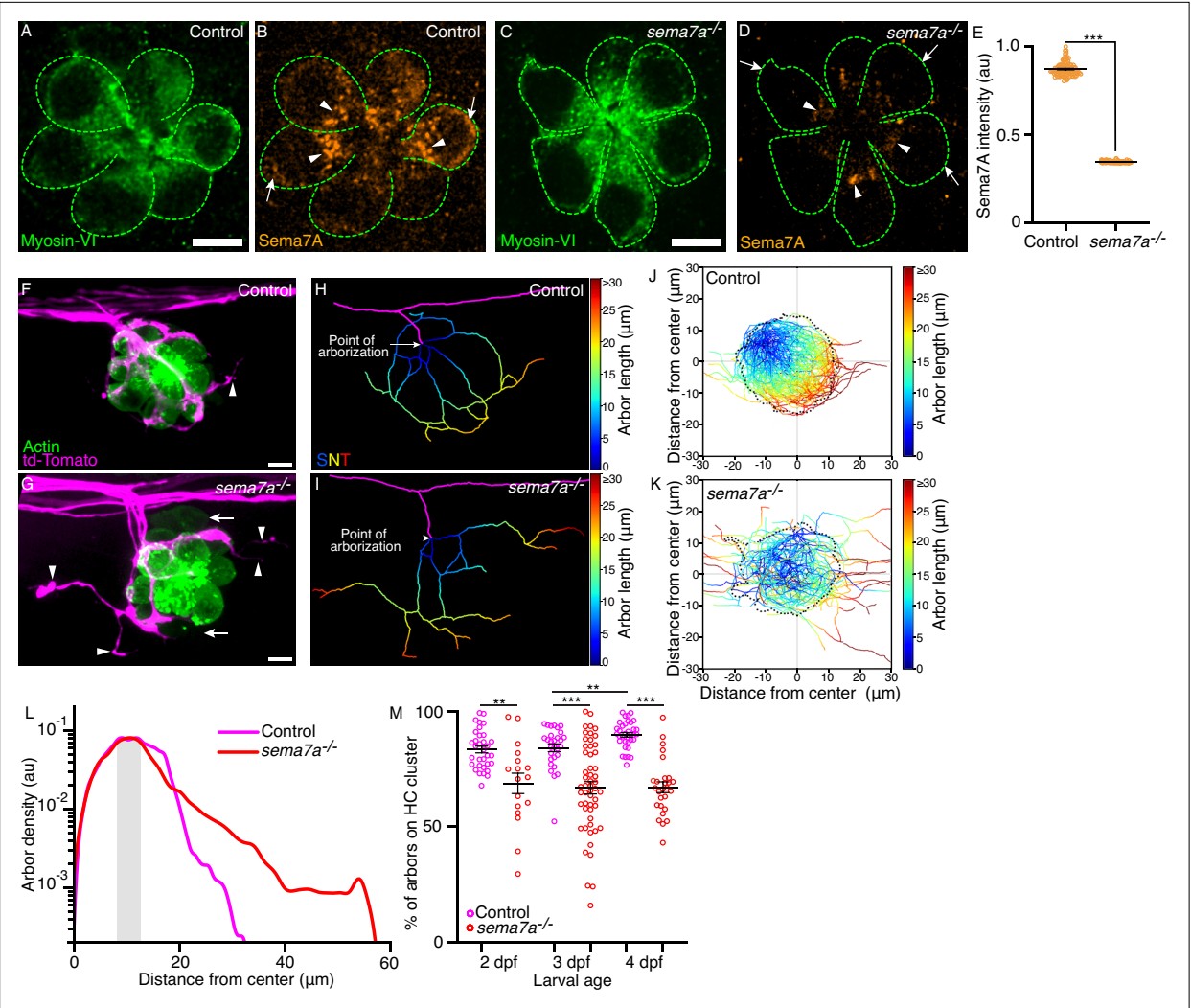

**Figure 2.** *sema7a⁻/⁻* mutants display aberrant sensory axon arborizations. (**A–D**) In micrographs of 3 dpf control and *sema7a⁻/⁻* neuromasts, Myosin VI (green) marks the hair cells (green dashed lines). The level of Sema7A (orange) is highly reduced in the *sema7a⁻/⁻* mutant compared to the control, with sporadic localization in the subapical region (arrowheads) and none in the basolateral (arrow) region. (**E**) A plot of normalized Sema7A intensity from 99 control and 100 *sema7a⁻/⁻* hair cells quantitates the effect. (**F, G**) Surface views of a control and a *sema7a⁻/⁻* neuromast at 4 dpf depict the interaction of the sensory arbors (magenta) with hair cell clusters (green). In the control, the arbors intimately contact the hair cells, with a few exceptions (arrowhead). In the *sema7a⁻/⁻* mutant, the arbors direct many aberrant projections (arrowheads) away from the hair cell cluster. The two immature hair cell pairs in the *sema7a⁻/⁻* neuromast are indicated by arrows. (**H,I**) Skeletonized networks portray the three-dimensional topology of the sensory arbors from the control and the *sema7a⁻/⁻* neuromasts depicted in panels F and G, respectively. The pseudocolored trajectories depict the increase in arbor contour length from each point of arborization, defined as the point at which the lateral line branch (magenta) contacts hair cell cluster. (**J, K**) Micrographs depict the skeletonized networks of the combined 4 dpf hair cell clusters, whose centers are located at (0,0). The X- and Y-coordinates represent the anteroposterior (AP) and the dorsoventral (DV) axes of the larva, respectively. Positive values of the X- and Y-coordinates represent the posterior and ventral directions, respectively. Combined skeletonized network traces from 27 control and 27 *sema7a⁻/⁻* mutant neuromasts are represented. (**L**) The plot denotes the densities of the sensory arbors around the center of the combined hair cell clusters for 35 control (magenta) and 27 *sema7a⁻/⁻* (red) neuromasts at 4 dpf. The shaded area marks the region proximal to the boundary of the combined hair cell cluster. (**M**) The plot quantifies the degree of contact of the sensory arbors to their hair cell clusters in individual neuromasts from both control (magenta) and *sema7a⁻/⁻* mutants (red), each point represents a single neuromast. Thirty-three, 29, and 35 neuromasts were analyzed from 2 dpf, 3 dpf, and 4 dpf control larvae, respectively. Seventeen, 53, and 27 neuromasts were analyzed from 2 dpf, 3 dpf, and 4 dpf *sema7a⁻/⁻* mutant larvae, respectively. HC, hair cell; Scale bars, 5 μm; au, arbitrary unit; means ± SEMs; *** signifies p<0.001; ** signifies p<0.01.

The online version of this article includes the following source data and figure supplement(s) for figure 2:

**Source data 1.** The source data contains comparisn of hair cell numbers, Sema7A intensity, and FM4-64 intensities between the control and the mutant larvae.

**Figure supplement 1.** *sema7a⁻/⁻* mutants exhibit aberrant sensory-axon arborization throughout development.

**Figure supplement 2.** Quantification of sensory arbor distributions and contacts with hair cell clusters.

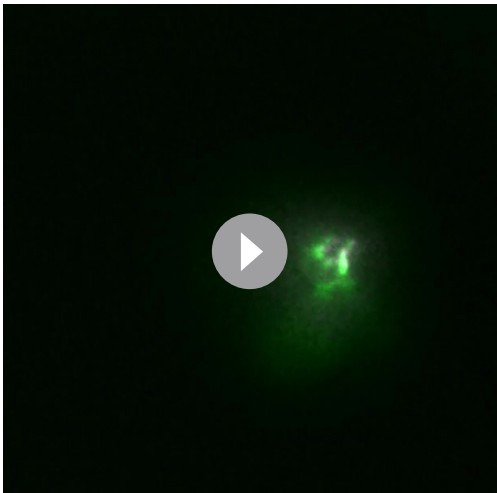

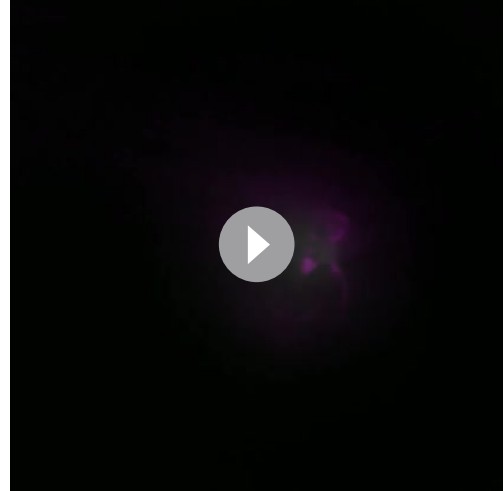

**Video 2.** FM4-64 labels the hair cells of both control and *sema7a*⁻/⁻ mutant neuromasts. Through-focus scans of 4 dpf control (*Video 2*) and *sema7a*⁻/⁻ mutant (*Video 3*) neuromasts depict the hair cells (green) that have incorporated FM4-64. The sensory arbors (magenta) are tightly associated with the clustered hair cells in the control neuromast but show aberrant arborization in the *sema7a*⁻/⁻ larvae.

https://elifesciences.org/articles/89926/figures#video2

**Video 3.** FM4-64 labels the hair cells of both control and *sema7a*⁻/⁻ mutant neuromasts. Through-focus scans of 4 dpf control (*Video 2*) and *sema7a*⁻/⁻ mutant (*Video 3*) neuromasts depict the hair cells (green) that have incorporated FM4-64. The sensory arbors (magenta) are tightly associated with the clustered hair cells in the control neuromast but show aberrant arborization in the *sema7a*⁻/⁻ larvae.

https://elifesciences.org/articles/89926/figures#video3

with simultaneous disruption of both signaling modalities of Sema7A, the organ fails to restrict the localization of sensory arbors and has significantly diminished contact with them.

## Sema7A patterns the arborization network's topology

Accurate spatial arrangement of the neuronal projections around hair cells is essential for proper retrieval and transmission of sensory information. Mutations that perturb proper assembly of sensory neuronal microcircuits lead to hearing impairment (*Diaz-Horta et al., 2016*). Quantitative insight into the wiring pattern is therefore essential to identify systematic aberrations in the microcircuit arising from genetic deficits. The three-dimensional skeletonized network traces allowed us to accurately delineate the microcircuit topology across individuals, developmental stages, and genotypes from living animals (*Figure 2D and E*). To quantitatively analyze the arbor network, we defined five topological attributes: (*Lykissas et al., 2007*) nodes, denoted as junctions where arbors branch, converge, or cross; (*Vaccaro et al., 2022*) loops, comprising minimal arbor cycles forming topological holes; (*Nguyen-Ba-Charvet et al., 2008*) bare terminals, defined as arbors with free ends; (*Jeroen Pasterkamp et al., 2003*) the node degree, calculated as the number of first-neighbor nodes incident to a node; and (*Nagiel et al., 2008*) the arbor curvature, measured as the total curvature of a bare terminal (*Figure 3A*). We quantified these attributes to accurately determine how the circuit topology is established during maturation of the sensory organ in control larvae and how it is impacted by *sema7a* deficit.

To determine how internode connectivity is distributed across networks, we quantified the distribution of node degrees along the relative height of the hair cell cluster across individuals, developmental stages, and genotypes. As for most other real-world networks (*Newman, 2018*), we observed a highly skewed node-degree

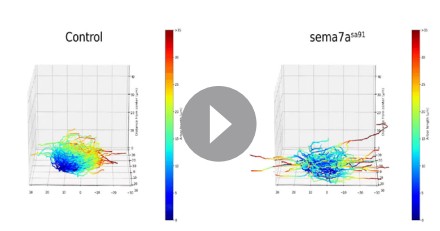

**Video 4.** Three-dimensional arborization patterns in control and *sema7a*⁻/⁻ neuromasts. Lateral views depict three-dimensionally rendered combined skeletonized network traces from 27 control and 27 *sema7a*⁻/⁻ mutant neuromasts at 4 dpf.

https://elifesciences.org/articles/89926/figures#video4

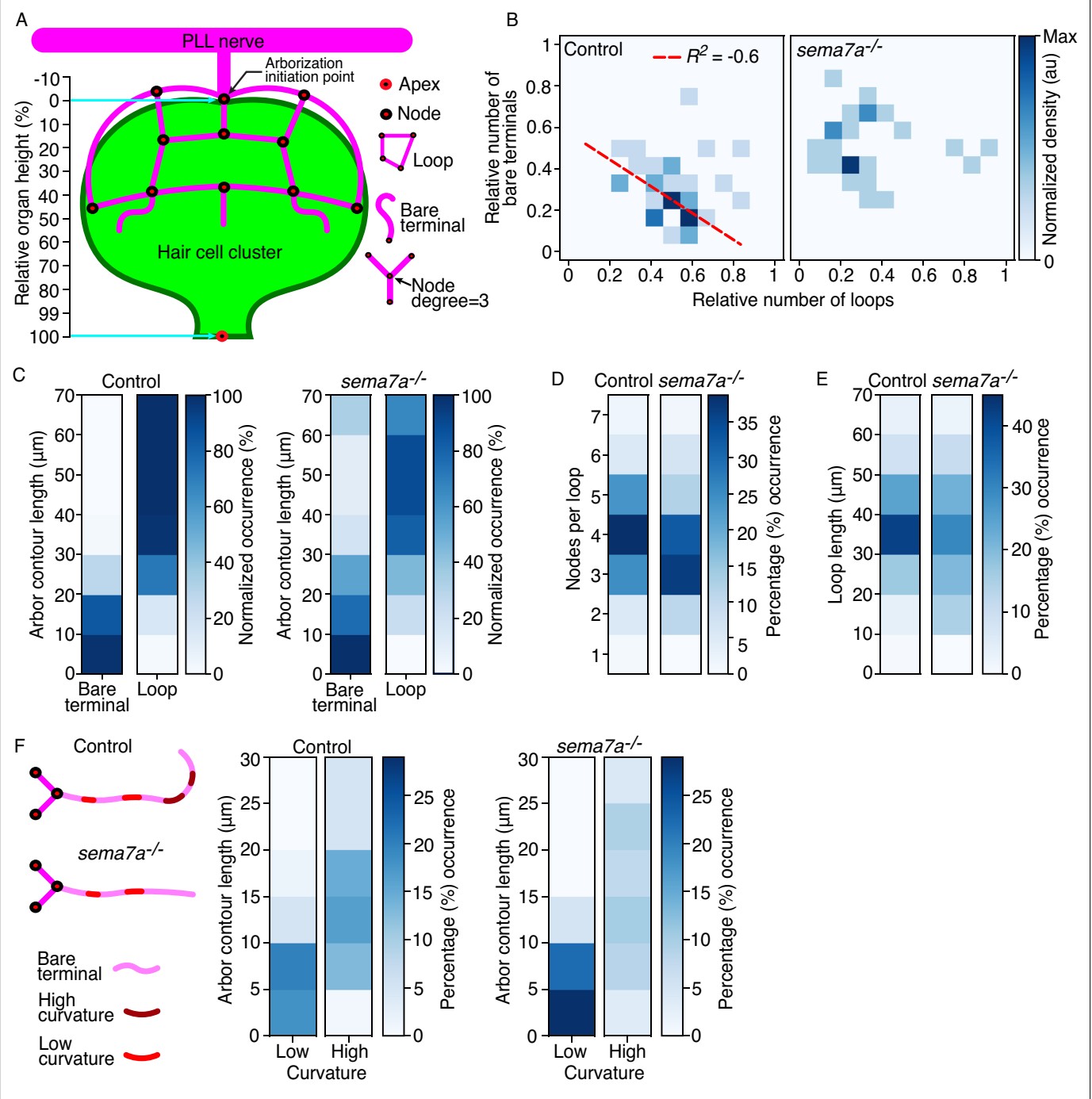

**Figure 3.** Sema7A patterns the topology of the sensory arbor network. (**A**) A schematic drawing of the combined hair cell cluster (green) and associated sensory axons (magenta) shows diverse topological features of the arborization network, including nodes, loops, bare terminals, node degree, and bare-terminal curvature. The apex is defined as the center of the hair bundle cluster and the base is denoted by the node from which the arborizations extends. The relative organ height describes the position along the axis formed between the arborization initiation point and the apex of the hair cell cluster. (**B**) The correlation between the relative number of bare terminals and loops from 35 control neuromasts and 27 *sema7a*$^{-/-}$ neuromasts at 4 dpf shows a negative correlation in the control but a scattered distribution in the mutants. The same neuromasts were analyzed in the subsequent panels. $R^2$, correlation coefficient. Dashed red line depicts the regression slope. (**C**) The relationship of contour lengths to bare terminals and loops reveals reduced looping in the mutants. (**D**) Quantification of the number of nodes per loop demonstrates a decrease in mutants. (**E**) Mutant larvae manifest a broader distribution of loop lengths compared to controls. (**F**) A schematic diagram of the extending bare terminals depicts regions of high and low curvature along the length of the arbor. The distributions of bare terminals between low- and high-curvature groups show an enrichment of long, highly curved arbors in the control and short, linear arbors in the mutants. Statistical analyses are displayed in *Table 1*.

*Figure 3 continued on next page*

*Figure 3 continued*

The online version of this article includes the following figure supplement(s) for figure 3:

**Figure supplement 1.** Quantification of topological features across developmental stages and genotypes.

**Figure supplement 2.** Quantification of topological features between control and *sema7a⁻ᐟ⁻* larvae across developmental stages.

distribution: the majority of nodes had low node degrees (three and four) and a small number of nodes—often acting as hubs, like the arborization initiation point—had high node degrees (five to seven). The node degree varied from three to seven in control larvae and three to six in *sema7a⁻ᐟ⁻* larvae. Across developmental stages and genotypes, the high node degrees primarily occurred at the base (region from 0 % to 10 % of the cluster height) of the hair cell cluster and acted as hubs for the majority of branching events. Low node degrees primarily occurred along the basolateral region (region from 10 % to 60 % of the cluster height) of the hair cell cluster. The apical region (region from 60 % to 100 % of the cluster height) was sparsely populated with low node degrees. The network's hierarchical organization along the organ's axis suggests that arbors receive sensory inputs at multiple levels, allowing the network to carry out robust neural computations. Even though nodes of low degree were most prevalent between genotypes across developmental stages, their distribution along the relative organ axis became significantly different at 4 dpf, with nodes in control larvae localized higher along the axis than in *sema7a⁻ᐟ⁻* larvae (*Figure 3A*; *Figure 3—figure supplement 1A–C*; *Table 1*). These observations indicate that Sema7A regulates both the spatial organization and node-degree complexity of the sensory arbor network.

**Table 1.** Statistical comparison between control and *sema7a⁻ᐟ⁻* larvae for topological features. One-sided Kolmogorov-Smirnov tests were used to assess the difference in node degree heights, nodes per loop, and loop lengths for control *versus sema7a⁻ᐟ⁻* neuromasts at 2 dpf, 3 dpf, and 4 dpf. In each of these cases, the difference in the distributions were not significantly different (p-values in red) between the control and the mutants at 2 dpf and 3 dpf but became significantly different (p-values in blue) at 4 dpf. Chi-square tests were used to assess the difference in occurrences of bare terminals versus loops, as well as the occurrences of bare terminals with low versus high curvature. The distribution at 4 dpf shows significant difference between the control and the mutant larvae. The occurrence of significant differences at 4 dpf for diverse topological features suggest that Sema7A function is required to achieve an ordered network topology at a mature stage. Thirty-three, 29, and 35 neuromasts were analyzed from 2 dpf, 3 dpf, and 4 dpf control larvae, respectively, and 17, 53, and 27 neuromasts were analyzed from 2 dpf, 3 dpf, and 4 dpf *sema7a⁻ᐟ⁻* mutant larvae, respectively. * Signifies p<0.05, ** signifies p<0.01, and *** signifies p<0.001.

| | p-values (control vs *sema7a⁻ᐟ⁻*) | | |
| --- | --- | --- | --- |
| **Node degree** | **2 dpf** | **3 dpf** | **4 dpf** |
| Degree = 3 | 3.50E-01 | 1.05E-08*** | 1.68E-03** |
| Degree = 4 | 3.85E-01 | 7.11E-02 | 1.71E-03** |
| Degree = 5 | 4.60E-01 | 5.23E-01 | 4.40E-01 |
| Bare terminals vs loop | 2 dpf | 3 dpf | 4 dpf |
| 0–10 µm | 3.84E-01 | 7.89E-01 | 2.11E-01 |
| 10–20 µm | 3.50E-02* | 3.92E-01 | 2.94E-02* |
| 20–30 µm | 2.28E-04*** | 2.63E-02* | 3.82E-09*** |
| 30–40 µm | 5.04E-059*** | 3.63E-25*** | 1.28E-33*** |
| 40–70 µm | --- | --- | --- |
| Nodes per loop | 5.06E-01 | 1.96E-01 | 3.50E-02* |
| Loop length | 5.81E-01 | 2.71E-01 | 2.21E-02* |
| | 2 dpf | 3 dpf | 4 dpf |
| Distribution of bare-terminal curvature | 4.23E-01 | 8.91E-01 | 1.99E-02* |

An average of 9.3 sensory axonal branches approach the hair cell cluster of a neuromast (*Dow et al., 2018*). Pathfinding of sensory axons around the hair cell cluster results in the formation of axonal fascicles that form numerous loops and bare terminals, which determine the microcircuit topology (*Figure 2B and D*; *Dow et al., 2018*). To optimally surround and contact the hair cell cluster, do the sensory arbors prefer to adopt one topological feature over the other? To address this question, we measured the correlation between the number of bare terminals and the number of loops across individuals, developmental stages, and genotypes. In both 2 dpf and 3 dpf larvae, the distributions remained highly scattered across individuals and genotypes. However, by 4 dpf in the control larvae the arbor network contained either a high number of loops with a low number of bare terminals or a high number of bare terminals with a low number of loops. This result suggests that the stereotypical circuit topology observed in the mature organ may emerge through transition of individual arbors from forming bare terminals to forming closed loops encircling topological holes. That we did not observe a similar transition in *sema7a*$^{-/-}$ neuromasts of the same age suggests that Sema7A may direct the switch between these two distinct topological features (*Figure 3B*; *Figure 3—figure supplement 1D, E*; *Table 1*).

Measurement of the contour length of both bare terminals and loops additionally revealed that across developmental stages in control larvae the sensory arbors followed a three-step process to choose the appropriate topological feature: arbors shorter than 10 µm always formed bare terminals; arbors between 10 µm and 30 µm began to switch between features, with arbors longer than 20 µm strongly preferring to form loops; and arbors longer than 30 µm always formed loops. These results imply that the switch between bare terminals and loops is partly dependent on the length of the arbors. However, this characteristic topological preference was significantly perturbed in the *sema7a*$^{-/-}$ larvae across developmental stages, as many of the sensory arbors longer than 30 µm remained as bare terminals (*Figure 3C*; *Figure 3—figure supplement 1F and G*; *Table 1*). Furthermore, in 4 dpf control neuromasts the majority of the loops contained four nodes and utilized 30–40 µm of arbors to form closed cycles. These conserved topological attributes were significantly perturbed in the *sema7a*$^{-/-}$ larvae of same age (*Figure 3D and E*; *Figure 3—figure supplement 2A–D*; *Table 1*). These findings indicate that Sema7A tightly monitors the sensory arbor loop formation, including the preferred loop length and number of nodes per loop, to properly surround and contact the hair cell cluster.

As observed previously, bare terminals in control neuromasts project transverse to the organ and linger nearby, often curving back to form new loops and contact the hair cell cluster (*Figure 2D and F*). To analyze the propensity of the bare terminals to curl backward, we measured their total curvatures, defined as the sum of the curvature at each point along the length of an arbor. Henceforth, we address the total curvature as the curvature of a bare terminal. We classified the bare terminals in two categories: high-curvature arbors with curled morphology and curvature greater than $\pi/6$ (30 degrees); and low-curvature arbors with linear morphology and curvature less than $\pi/6$. In control neuromasts 44.8% of arbors had low curvature compared to 56.4% in *sema7a*$^{-/-}$ neuromasts at 4 dpf. Concomitantly, 55.2% of arbors had high curvature in control neuromasts compared to 43.6% in *sema7a*$^{-/-}$ neuromasts at 4 dpf (*Figure 3F*; *Figure 3—figure supplement 2E, F*; *Table 1*). These observations further confirm our hypothesis that a Sema7A-mediated attractive cue is involved in orienting bare neural arbors toward the hair cell cluster to potentially form loops.

In summary, we propose that Sema7A mediates four key aspects of the topology of the arborization network: (1) the node degree complexity along the organ's axis; (2) the transition of arbors from bare terminals to loops; (3) the preference for forming loops containing four nodes with 30–40 µm of arbors; and (4) the preference of bare terminals to adopt higher curvature.

## Sema7A$^{sec}$ is a sufficient chemoattractive cue for sensory axon guidance

Because genetic inactivation of the *sema7a* gene disrupts both signaling modalities of the cognate protein, we could not assess the specific activity of the Sema7A$^{sec}$ diffusive cue in guiding lateral-line sensory axons. To independently verify the potential role of that isoform, we therefore expressed the secreted variant ectopically and investigated the resultant arborization patterns.

Analysis of the microcircuit connectivity of the neuromasts has demonstrated bare sensory-axonal terminals in the perisynaptic compartments, where they do not contact the hair cell membrane (*Dow et al., 2018*). We speculate that these bare terminals are attracted by a Sema7A$^{sec}$ diffusive cue.

To test this hypothesis, we ectopically expressed Sema7A$^{sec}$ protein tagged with the fluorophore mKate2, then analyzed its effect on the morphology of sensory arbors. Fertilized one-cell embryos were injected with the *hsp70:sema7a$^{sec}$-mKate2* plasmid and raised to 3 dpf. Larvae expressing the transgenesis marker—the lens-specific red fluorescent protein mCherry—were heat-shocked, incubated to allow expression, and subsequently mounted for live imaging. In these larvae, a random mosaic of cells—often embryonic muscle progenitors in the dermomyotome or mature myofibers—expressed the exogenous Sema7A$^{sec}$-mKate2 protein. We selected only those neuromasts in which a single mosaic ectopic integration had occurred near the network of sensory arbors (*Figure 4A and B*). In all 22 such cases, we observed robust axonal projections from the sensory arbor network toward the ectopically expressing Sema7A$^{sec}$ cells. When the exogenous *sema7a$^{sec}$* integration occurred in embryonic muscle progenitor cells, which reside in a superficial layer external to the muscle fibers and adjacent to the larval skin (*Sharma et al., 2019*), the projections were able to form direct contacts with them (*Figure 4C*; *Videos 5 and 6*). But more often, when mature myofibers expressed the exogenous Sema7A$^{sec}$ deep in the myotome, the axonal projections approached those targets but failed to contact them (*Figure 4D*, *Videos 7 and 8*). This behavior likely arose because other components in the myotomal niche—including the fibrous epimysium, other myofibers, myoblasts, and immature myotubes whose surroundings are permeable to diverse diffusive cues (*Keenan and Currie, 2019*)—physically obstructed direct interaction of the extended neurites with the ectopically expressing myofibers. All 18 of the injected but not heat-shocked control larvae did not express ectopic Sema7A$^{sec}$, and we did not observe aberrant projections from the sensory arbor network (*Figure 4E*).

To measure the accuracy of the extended axonal projections in finding the ectopic Sema7A$^{sec}$ source, we defined three parameters: (1) the source path, denoted as the linear distance from the point of arborization to the boundary of an ectopically expressing Sema7A$^{sec}$ cell; (2) the projection path, representing the linear distance from the point of arborization to the terminus of the extended projection; and (3) the projection proximity, calculated as the linear distance from the terminus of the extended axonal projection to the nearest boundary of the ectopically expressing Sema7A$^{sec}$ cell (*Figure 4B*). We quantified these parameters from 18 of the 22 mosaic integration events; in the remainder the extended axonal projections–although following the ectopic source–reentered the posterior lateral-line nerve so that identification of the axon terminals was not possible (*Figure 4— figure supplement 1A–A″*; *Video 9*). The projection paths of the extended axons reached from 27.1 µm to 142.5 µm while following the ectopic Sema7A$^{sec}$ cues. When we plotted the projection path length against the source-path length, we observed a nearly perfect correlation (*Figure 4F*). This result signifies that irrespective of the location of the ectopic source–whether proximal or distal to the sensory arbor network–the Sema7A$^{sec}$ cue sufficed to attract axonal terminals from the sensory arbor. Furthermore, the low average projection-proximity length of 3.5±0.7 µm (mean ± SEM) indicated that the extended projection terminals remained either in contact (8 of 18) or in the vicinity (10 of 18) of the ectopic Sema7A$^{sec}$ sources (*Figure 4G*). These observations together demonstrate that the Sema7A$^{sec}$ diffusive cue is sufficient to provide neural guidance in vivo.

As the posterior lateral-line ganglion matures, newly formed neurons extend their axonal growth cones along the posterior lateral-line nerve to reach the neuromasts (*Sato et al., 2010*). We wondered whether the axonal projections that had yet to be associated with a neuromast could also respond to an ectopic source of Sema7A$^{sec}$. Indeed, on seven occasions in which *sema7a$^{sec}$* integration had occurred between two neuromasts, we detected neurite extensions from the posterior lateral-line nerve to the ectopic Sema7A$^{sec}$ sources (*Figure 4H*; *Figure 4—figure supplement 1B–B″*, *Videos 10–12*). Live time-lapse video microscopy further revealed dynamic association between the extending neurites and the ectopic Sema7A$^{sec}$ sources. Long and thin along their entire extents, the neurites were highly motile, extending and retracting rapidly near the Sema7A$^{sec}$ expressing cells. Many neurites maintained their contact with the ectopic sources of Sema7A$^{sec}$ from one hour to almost eight hours (*Figure 4I and J*; *Figure 4—figure supplement 2A*, *Videos 13–16*). The responsiveness of the posterior lateral-line axons to Sema7A$^{sec}$ might therefore be an intrinsic property of the neurons that does not require the neuromast microenvironment.

## Sema7A-GPI fails to impart sensory axon guidance from a distance

We speculate that the GPI-anchored transcript variant of the *sema7a* gene can function only through contact-dependent pathways. To test this hypothesis, we expressed the Sema7A-GPI ectopically and

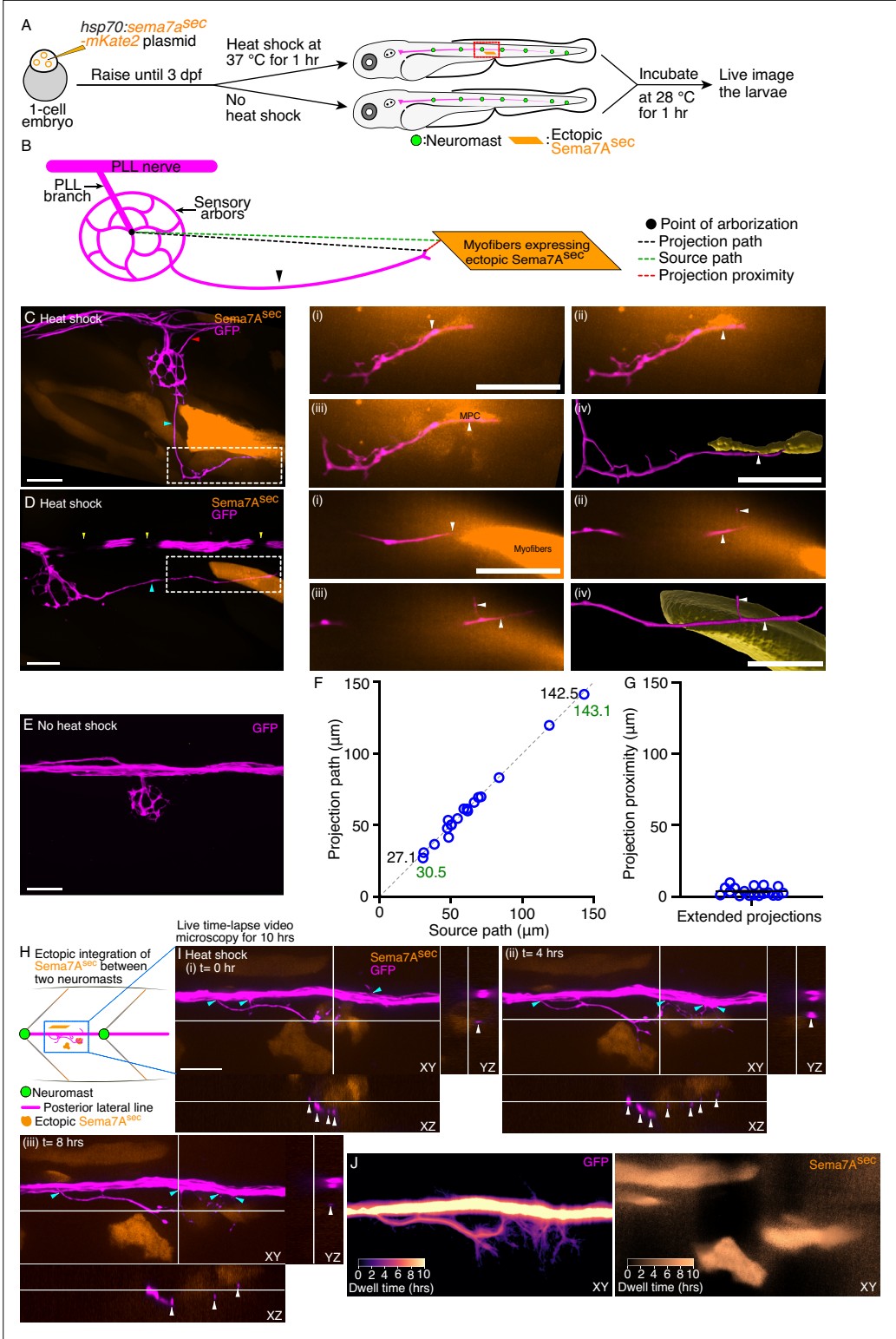

**Figure 4.** Ectopic Sema7A^sec diffusive cue provides neural guidance in vivo. (**A**) A diagrammatic overview depicts the generation of a transgenic animal that expresses the Sema7A^sec protein ectopically under the control of a thermally inducible promoter. Larvae with ectopic myotomal-integration near the network of sensory arbors (red rectangle) were imaged to analyze arbor morphology. (**B**) A schematic drawing of a sensory arbor from a heat-shocked larva depicts an extended axonal projection (arrowhead) that reaches toward the myofibers expressing the ectopic Sema7A^sec protein. Parameters that quantitate the accuracy of the extended axonal projections toward

*Figure 4 continued*

the ectopic Sema7A$^{sec}$ sources are denoted. (**C**) In a micrograph of an ectopically expressing Sema7A$^{sec}$ (orange) larva, the sensory arbor (magenta) extends two aberrant axonal projections. One elongates (cyan arrowhead) along the somite boundary to reach and contact an ectopically integrated muscle progenitor cell (white dashed line) and the other (red arrowhead) reenters the posterior lateral-line nerve while following a second ectopic source. The through-focus scans (i–iii) from the epidermis to the dermomyotome and the three-dimensional (3D) surface reconstruction (iv) reveal the intimate contact between the aberrant sensory arbor (arrowheads) and the muscle progenitor cell. (**D**) In a micrograph of an ectopically expressing Sema7A$^{sec}$ (orange) larva, the sensory arbor (magenta) extends a single aberrant axonal process (cyan arrowhead) to reach ectopically integrated myofibers (white dashed line). The through-focus scans (i–iii) from the epidermis to the dermomyotome and the 3D surface reconstruction (iv) reveal the proximal association of the aberrant sensory arbor (arrowheads) to the myofibers. Melanocytes (yellow arrowhead) along the horizontal myoseptum intermittently block the visibility of the lateral-line nerve. (**E**) An injected, but not heat-shocked, control larva does not express ectopic Sema7A$^{sec}$ and does not show aberrant projection from the sensory arbor. (**F**) A plot demonstrates the accuracy of 18 extended axonal arbors in finding ectopic Sema7A$^{sec}$ sources. Each circle represents a single ectopic integration event. The two pairs of numbers represent the minimal and maximal lengths of the projection path (black) and its corresponding source path (green). (**G**) A plot quantitates the distribution of projection-proximity length from 18 ectopic integration events. (**H**) A schematic drawing of a section of the lateral-line nerve between two neuromasts from a heat-shocked larva depicts a few extended neurites (magenta) reaching toward the cells expressing the ectopic Sema7A$^{sec}$ protein (orange). The Sema7A$^{sec}$ integration site (blue rectangle) with the sensory axonal projections was imaged for ten hours to visualize axonal dynamics. (**I**) Micrographs (i-iii) at three distinct times in the time-lapse video microscopy show directed branching (cyan arrowheads) of the sensory neurites from the lateral-line nerve toward the ectopic Sema7A$^{sec}$ sources. Each panel depicts the ectopic integration site and the associated sensory neurites from three different planes, XY, XZ, and YZ. White arrowheads in XZ and YZ planes highlight the close association of the sensory axons with the ectopic Sema7A$^{sec}$ sources. The t=0 hr denotes the onset of imaging, which is two hours post the beginning of heat shock. (**J**) The micrographs depict in pseudocolored trajectories the dwell times of the sensory neurites (left) at the Sema7A$^{sec}$ sources (right) for ten hours. MPC, muscle progenitor cell; scale bars, 20 µm; means ± SEMs.

The online version of this article includes the following source data and figure supplement(s) for figure 4:

**Source data 1.** The source data depicts the accuracy of the 18 extended axonal arbors in finding ectopic Sema7Asec sources.

**Figure supplement 1.** Ectopic Sema7A$^{sec}$ diffusive cues induce aberrant sensory neurite extension.

**Figure supplement 2.** Ectopic integration of Sema7A$^{sec}$ distant from the neuromast induces sensory neurite extension.

investigated the resultant arborization patterns of the sensory axons. Fertilized one-cell embryos were injected with the *hsp70:sema7a-GPI-2A-mCherry* plasmid and raised to 3 dpf. Larvae expressing the transgenesis marker—the lens-specific green fluorescent protein (GFP)—were heat-shocked, incubated to allow expression, and subsequently mounted for live imaging (*Figure 5A*). In these larvae, a few mature myofibers expressed the exogenous Sema7A-GPI protein. We selected

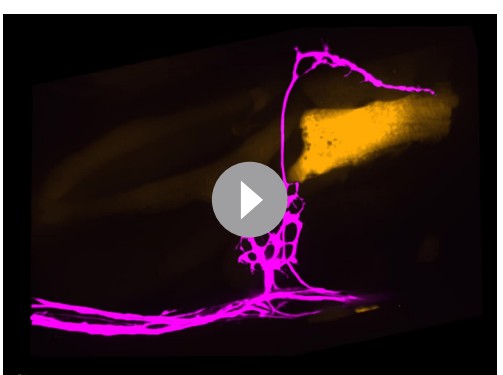

**Video 5.** Sensory axonal projections contact an embryonic muscle progenitor cell expressing Sema7A$^{sec}$. A through-focus scan of a 3 dpf larva depicts a muscle progenitor cell in the dermomyotome that expresses the ectopic Sema7A$^{sec}$ (orange) and the sensory axonal projection (magenta) that makes intimate contact with it. Scale bar, 20 µm.
https://elifesciences.org/articles/89926/figures#video5

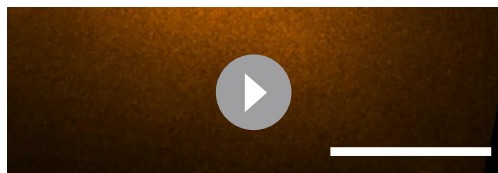

**Video 6.** Volumetric surface reconstruction of a sensory arbor and an embryonic muscle progenitor expressing Sema7A$^{sec}$. Reconstruction of the muscle progenitor cell (orange) at 3 dpf reveals its intimate association with an attracted sensory arbor (magenta).
https://elifesciences.org/articles/89926/figures#video6

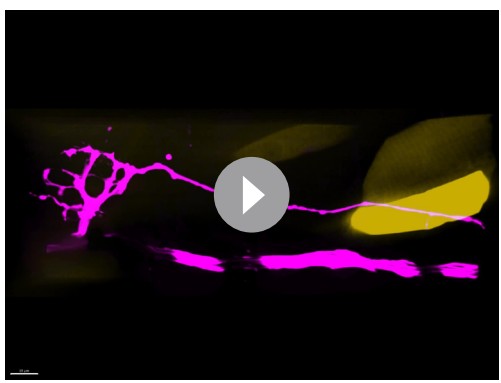

**Video 7.** Volumetric surface reconstruction of a sensory arbor and the mature myofibers expressing Sema7A^sec. Reconstruction of mature myofibers expressing ectopic Sema7A^sec (orange) at 3 dpf shows their close association with the attracted sensory arbors (magenta). https://elifesciences.org/articles/89926/figures#video7

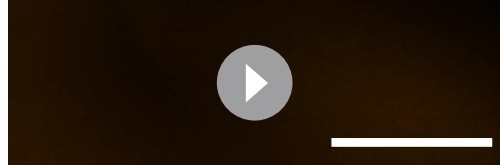

**Video 8.** Sensory axonal projections remain in proximity to myofibers expressing Sema7A^sec. A through-focus scan of a 3 dpf larva depicts mature myofibers expressing exogenous Sema7A^sec deep in the myotome (orange) and the sensory axonal projection (magenta) that remains nearby. https://elifesciences.org/articles/89926/figures#video8

only those neuromasts in which an ectopic integration had occurred near the network of sensory arbors. In all 15 such cases, no sensory axon left the arbor network to approach the ectopic Sema7A-GPI sources. In all 17 of the uninjected and heat-shocked control larvae, the sensory arbor network did not show any perturbation (*Figure 5B–C*, *Videos 17–19*). These data suggest that the Sema7A-GPI, unlike its secreted counterpart, functions at the surfaces of the expressing cells.

## Sema7A deficiency impacts synaptic assembly

Contact stabilization between an axon and its target cell is essential for synapse formation (*Shen and Cowan, 2010*). Neural GPI-anchored proteins can act as regulators of various synaptic-adhesion pathways (*Um and Ko, 2017*). In the mouse olfactory system, GPI-anchored Sema7A is enriched in olfactory sensory axons and mediates sustained interaction with the dendrites of mitral and tufted cells in the olfactory bulb to establish synapses (*Inoue et al., 2018*). In the developing mouse brain–particularly in Purkinje cells–GPI-anchored Sema7A instead regulates the elimination of synapses onto climbing fibers (*Uesaka et al., 2014*). The role of anchored Sema7A in regulating synaptic architecture can thus vary according to the cell type and developmental stage. Our demonstration that Sema7A is necessary to consolidate the contact between sensory axons and hair cells during neuromast development suggests a possible role in regulating synaptic structure.

Proper synaptogenesis requires the correct spatial organization of the presynaptic and postsynaptic apparatus at the apposition of two cells. At the interface between hair cells and sensory axons, the synaptic network involves two scaffolding proteins in particular: ribeye, a major constituent of the presynaptic ribbons (*Sheets et al., 2014*), and membrane-associated guanylate kinase (MAGUK), a conserved group of proteins that organize postsynaptic densities (*Oliva et al., 2012*). Utilizing these scaffolding proteins as markers, we investigated the impact of the *sema7a^-/-* mutation on synapse formation. To analyze the distribution of presynaptic densities–immunolabeled with an antibody against the ribeye-associated presynaptic constituent C-terminal binding protein (CTBP; *Sheets*

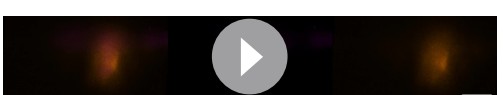

**Video 9.** Aberrant axonal projections reenter the lateral-line nerve while following an ectopic Sema7A^sec source. A through-focus scan of a 3 dpf larva depicts cells expressing exogenous Sema7A^sec (orange) that guides the aberrant sensory axonal projection (magenta) back into the lateral-line nerve. Scale bar, 20 μm. https://elifesciences.org/articles/89926/figures#video9

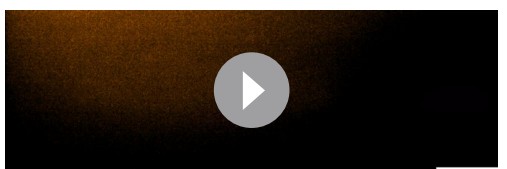

**Video 10.** Ectopic Sema7A^sec expression induces aberrant neurite projections from the lateral-line nerve. Through-focus scans of 3 dpf larvae depict three individual incidents of ectopic *sema7a^sec* (orange) integrations that occurred distant from the neuromasts. In each case, myofibers express ectopic Sema7A^sec protein (orange) that attracts neurite extensions from the lateral-line nerve (magenta). Scale bars, 20 μm. https://elifesciences.org/articles/89926/figures#video10

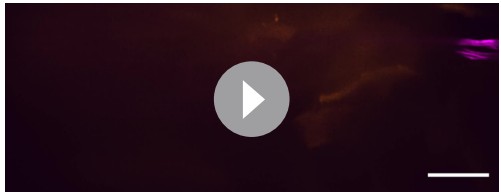

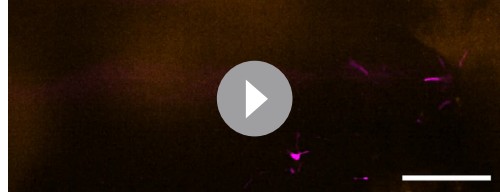

**Video 11.** Ectopic Sema7A^sec^ expression induces aberrant neurite projections from the lateral-line nerve. Through-focus scans of 3 dpf larvae depict three individual incidents of ectopic *sema7a*^sec^ (orange) integrations that occurred distant from the neuromasts. In each case, myofibers express ectopic Sema7A^sec^ protein (orange) that attracts neurite extensions from the lateral-line nerve (magenta). Scale bars, 20 μm.
https://elifesciences.org/articles/89926/figures#video11

**Video 12.** Ectopic Sema7A^sec^ expression induces aberrant neurite projections from the lateral-line nerve. Through-focus scans of 3 dpf larvae depict three individual incidents of ectopic *sema7a*^sec^ (orange) integrations that occurred distant from the neuromasts. In each case, myofibers express ectopic Sema7A^sec^ protein (orange) that attracts neurite extensions from the lateral-line nerve (magenta). Scale bars, 20 μm.
https://elifesciences.org/articles/89926/figures#video12

*et al., 2014*)–we counted the number and measured the area of CTBP punctae from multiple neuromasts at several stages of neuromast development (*Figure 6A and B*). The CTBP punctae in each hair cell ranged from zero to seven across developmental stages. In the zebrafish's posterior lateral line, the formation of presynaptic ribbons is an intrinsic property of the hair cells that does not require contact with lateral-line sensory axons. However, sustained contact with the sensory axon influences the maintenance and stability of ribbons (*Suli et al., 2016*). As neuromasts matured from 2 dpf to 4 dpf, we identified similar distribution patterns with a characteristic increase in the number of CTBP punctae in both control and *sema7a*^-/-^ larvae (*Figure 6C and D*). This result implies that the formation of new presynaptic ribbons is not perturbed in the mutants, even though there are on average fewer CTBP punctae (*Figure 6—figure supplement 1A–C*). Measurement of the areas of CTBP punctae also showed similar distributions in both genotypes (*Figure 6E and F*) but the average area of presynaptic densities was reduced in the mutants (*Figure 6—figure supplement 1D–F*, *Figure 6—figure supplement 2*). Because GPI-anchored Sema7A lacks a cytosolic domain, it is unlikely that Sema7A signaling directly induces the formation of presynaptic ribbons. We propose that the decrease in average

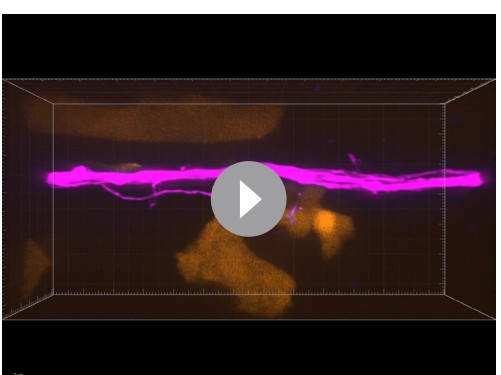

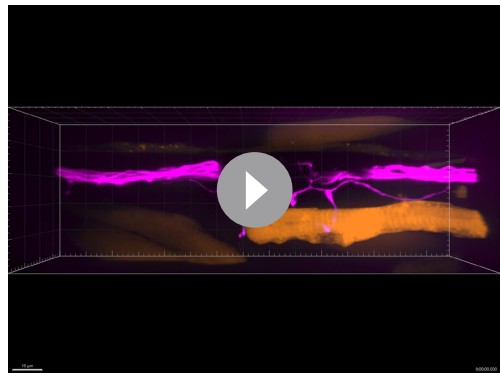

**Video 13.** Volumetric reconstructions depict close contacts between the sensory neurites and the ectopic Sema7a^sec^ expressing cells. Volumetric reconstructions of 3 dpf larvae depict two individual incidents of ectopic *sema7a*^sec^ (orange) integrations that occurred distant from the neuromasts. In each case, myofibers or random mosaic of cells in the dermomyotome express ectopic Sema7A^sec^ protein (orange) that attracts neurite extensions from the lateral-line nerve (magenta).
https://elifesciences.org/articles/89926/figures#video13

**Video 14.** Volumetric reconstructions depict close contacts between the sensory neurites and the ectopic Sema7a^sec^ expressing cells. Volumetric reconstructions of 3 dpf larvae depict two individual incidents of ectopic *sema7a*^sec^ (orange) integrations that occurred distant from the neuromasts. In each case, myofibers or random mosaic of cells in the dermomyotome express ectopic Sema7A^sec^ protein (orange) that attracts neurite extensions from the lateral-line nerve (magenta).
https://elifesciences.org/articles/89926/figures#video14

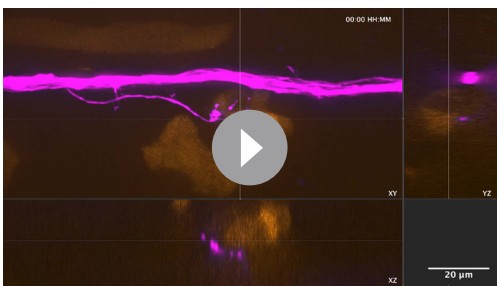

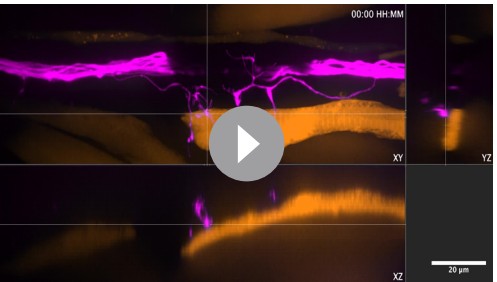

**Video 15.** Time-lapse video microscopies depict the dynamics between the sensory neurites and the ectopic Sema7a^sec expressing cells. Time-lapse images of 3 dpf larvae depict two individual incidents where the ectopic Sema7A^sec sources (orange) induce directed branching of the sensory neurites (magenta) from the lateral-line nerve toward itself. Within each time-lapse videos, individual panel depicts the ectopic integration site and the associated sensory neurites from three different planes, XY, XZ, and YZ. Scale bars, 20 μm.
https://elifesciences.org/articles/89926/figures#video15

**Video 16.** Time-lapse video microscopies depict the dynamics between the sensory neurites and the ectopic Sema7a^sec expressing cells. Time-lapse images of 3 dpf larvae depict two individual incidents where the ectopic Sema7A^sec sources (orange) induce directed branching of the sensory neurites (magenta) from the lateral-line nerve toward itself. Within each time-lapse videos, individual panel depicts the ectopic integration site and the associated sensory neurites from three different planes, XY, XZ, and YZ. Scale bars, 20 μm.
https://elifesciences.org/articles/89926/figures#video16

number and area of CTBP punctae likely reflects decreased stability of the presynaptic ribbons owing to lack of contact between the sensory axons and the hair cells.

Our analysis of the skeletonized network traces of the sensory arbor network identified distinct topological features that are resultant of collective assemblies of each sensory axon. Because postsynaptic aggregates are organized at the sensory axons, we wondered whether they are associated with a specific topological feature of the network. To identify the distribution of the postsynaptic aggregates along the arbor network, we utilized the transgenic animal *TgBAC(neurod1:EGFP)* that labeled the sensory axons of the posterior lateral line in green fluorescence protein (GFP), and marked the postsynaptic aggregates through immunostaining with a pan-MAGUK antibody (*Figure 6G*, *Video 20*).

We generated skeletonized network traces of the arbor network and determined the three-dimensional spatial locations of the MAGUK punctae (*Figure 6—figure supplement 2*). Comparison of the topological features of the network with the spatial locations of the MAGUK punctae revealed that the postsynaptic densities occur more extensively on the loops than on the bare terminals (*Figure 6G*, *Video 20*). We measured 567 MAGUK punctae from 16 sensory arbor networks, where 531 punctae were located on the loops and only 36 punctae on the bare terminals (*Figure 6H*). Furthermore, the MAGUK punctae predominantly aggregate near the node and their density sharply diminishes moving farther from the node (*Figure 6I*, *Figure 6—figure supplement 2*). These observations suggest that the distribution of the postsynaptic aggregates is closely associated with the topological attributes of the sensory circuit. Because in the *sema7a^-/-* mutants the arbor networks have significantly lower number of loops and nodes, and more bare terminals, we hypothesize that in the mutants the occurrence of postsynaptic aggregates is perturbed. Moreover, Sema7A-GPI-mediated juxtracrine signaling onto the apposed axonal membrane is essential to induce postsynaptic assembly (*Inoue et al., 2018*).

To characterize the impact of *sema7a* mutation on the formation of postsynaptic aggregates, we counted the number and measured the area of MAGUK punctae from multiple neuromasts at several stages of neuromast development (*Figure 6J and K*). At 4 dpf, 68.3% of the hair cells in the control neuromasts were associated with at least one MAGUK punctum. In the *sema7a^-/-* mutant this value fell to 37.5% (*Figure 6L and M*). The average number of MAGUK punctae also declined in the mutants (*Figure 6—figure supplement 1G–I*). As control neuromasts matured from 2 dpf to 4 dpf, the MAGUK punctae showed a shift toward smaller sizes, suggesting that the postsynaptic structure fragmented into smaller entities as observed in other excitatory synapses (*Taschenberger et al., 2002*). At 4 dpf, 77.5% of the hair cells in the control neuromasts had postsynaptic densities with areas between 0.10 μm$^2$ and 0.40 μm$^2$. In the *sema7a^-/-* mutant the corresponding value was only 42.5% (*Figure 6N and O*). The average area of MAGUK punctae was also reduced in the mutants

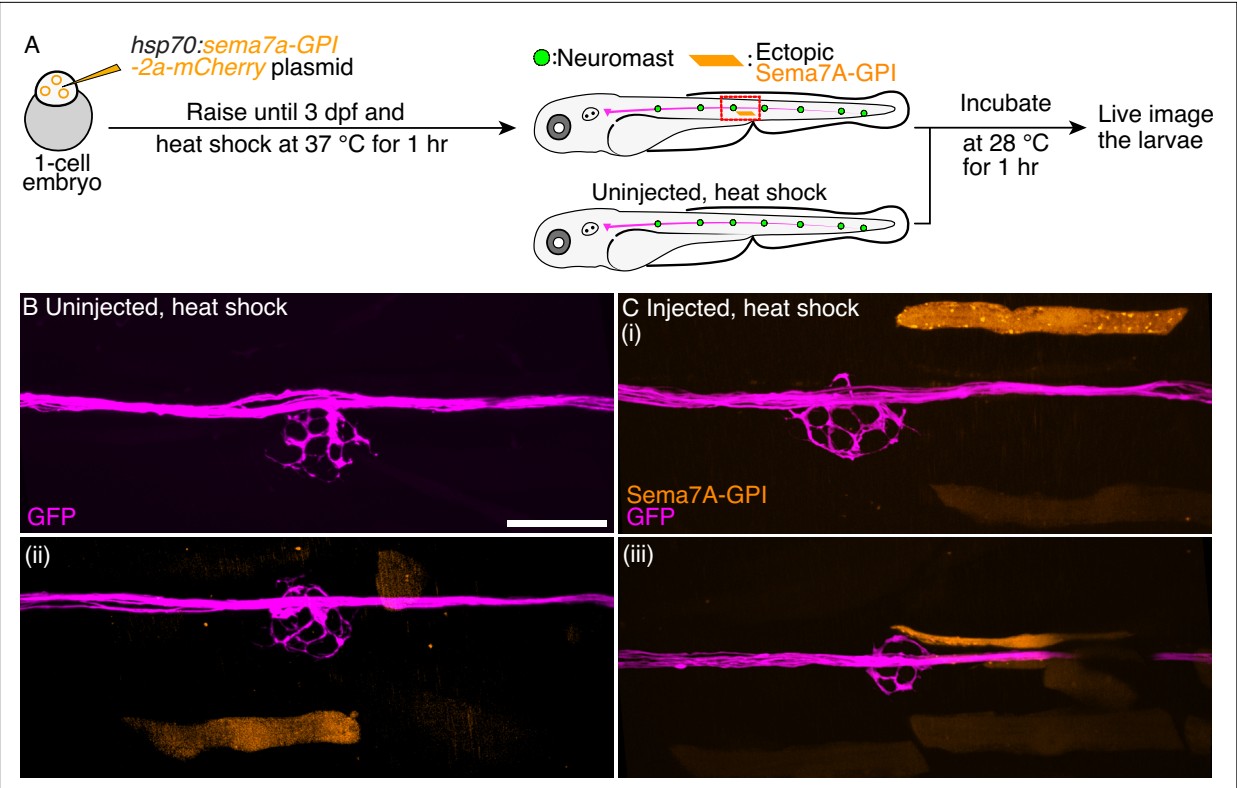

**Figure 5.** Ectopic Sema7A-GPI fails to guide sensory arbors from a distance. (**A**) A diagrammatic overview depicts the generation of a transgenic animal that expresses the Sema7A-GPI protein ectopically under the control of a thermally inducible promoter. Larvae with ectopic myotomal-integration near the network of sensory arbors (red rectangle) were imaged to analyze arbor morphology. Uninjected, heat-shocked larvae were used as controls. (**B**) Uninjected and heat-shocked larva depicts normal arborization pattern of the sensory axons (magenta). (**C**) Injected and heat-shocked larvae robustly express the Sema7A-GPI in the myofibers (orange). Panels (i-iii) depict the ectopically expressing Sema7A-GPI myofibers that fail to attract sensory arbors toward itself. Scale bar, 20 μm.

compared to the controls (*Figure 6—figure supplement 1J–L*, *Figure 6—figure supplement 2*). We speculate that the abnormalities in the postsynaptic structure of the *sema7a*$^{-/-}$ mutants arose from reduced contact between the hair cells and the sensory axons and perturbed topological attributes of the sensory circuit or from the lack of Sema7A-GPI-mediated juxtracrine signaling onto the apposed neurite terminals (*Inoue et al., 2018*).

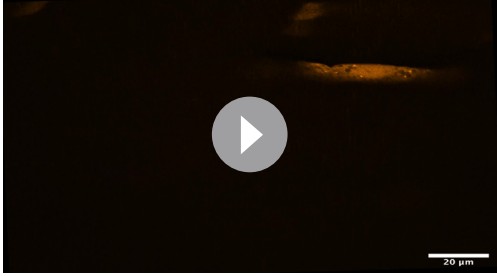

**Video 17.** Sema7A-GPI is incapable to impart sensory axon guidance from a distance. Through-focus scans of 3 dpf larvae depict three individual incidents of ectopic *sema7a-GPI* integrations that occurred near the sensory arbor network. In each case, myofibers robustly express ectopic Sema7A-GPI protein (orange) but the sensory arborization pattern (magenta) remain unperturbed. Scale bars, 20 μm.

https://elifesciences.org/articles/89926/figures#video17

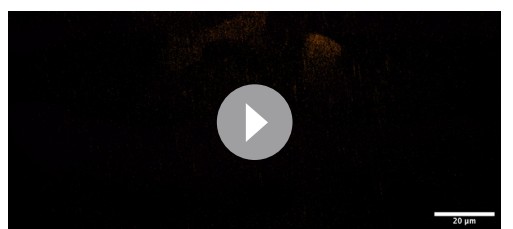

**Video 18.** Sema7A-GPI is incapable to impart sensory axon guidance from a distance. Through-focus scans of 3 dpf larvae depict three individual incidents of ectopic *sema7a-GPI* integrations that occurred near the sensory arbor network. In each case, myofibers robustly express ectopic Sema7A-GPI protein (orange) but the sensory arborization pattern (magenta) remain unperturbed. Scale bars, 20 μm.

https://elifesciences.org/articles/89926/figures#video18

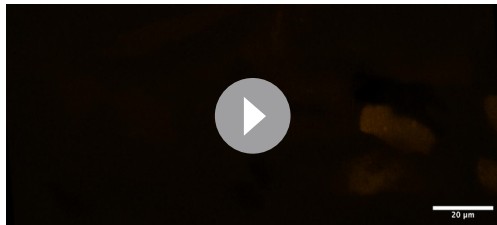

**Video 19.** Sema7A-GPI is incapable to impart sensory axon guidance from a distance. Through-focus scans of 3 dpf larvae depict three individual incidents of ectopic *sema7a-GPI* integrations that occurred near the sensory arbor network. In each case, myofibers robustly express ectopic Sema7A-GPI protein (orange) but the sensory arborization pattern (magenta) remain unperturbed. Scale bars, 20 µm.

https://elifesciences.org/articles/89926/figures#video19

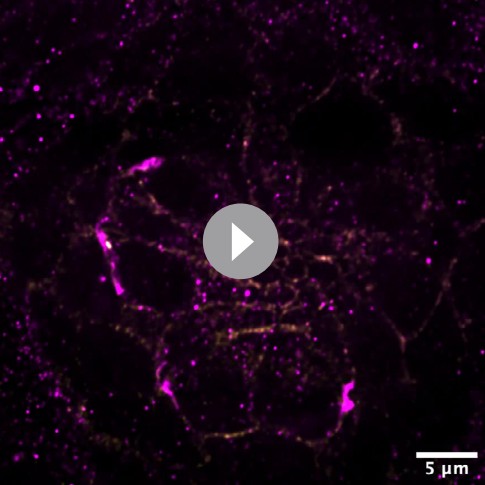

**Video 20.** Postsynaptic assemblies are enriched in the loops and near the nodes of a sensory arbor network. Through-focus scans of a 4 dpf neuromast depict the sensory arbor network (magenta) where the postsynaptic assemblies (yellow) are primarily located in the loops of the arbor network and are enriched near the nodes.

https://elifesciences.org/articles/89926/figures#video20

## Discussion

The specification of neural circuitry requires precise control of the guidance cues that direct and restrict neural arbors at their target fields and facilitate synapse formation. In the lateral-line neuromast of the zebrafish, we have identified Sema7A as an essential cue that regulates the interaction between hair cells and sensory axons. We have discovered two isoforms of the *sema7a* gene in vivo: the chemoattractive diffusible form is sufficient to guide the sensory arbors toward their target, whereas the membrane-attached form likely participates in sculpting accurate neural circuitry to facilitate contact-mediated formation and maintenance of synapses. Our results suggest a potential mechanism for hair cell innervation in which a local Sema7A$^{sec}$ diffusive cue likely consolidates the sensory arbors at the hair cell cluster and the membrane-anchored Sema7A-GPI molecule guides microcircuit topology and synapse assembly. Establishing the exact function of each of the *sema7a* isoforms will necessitate studies that include transcript-specific targeted mutagenesis and rescue of the *sema7a$^{-/-}$* mutant phenotype by each of the transcript variants.

The role of Sema7A in regulating diverse topological features of the arborization network suggests that microcircuits of control and *sema7a$^{-/-}$* neuromasts have important functional differences. Loss of Sema7A disrupts the hierarchical organization of the node-degree distribution along the organ's axis, as well as the preference towards forming loops containing four nodes with 30–40 µm of arbors. What benefit does the network gain from surrounding the hair cell cluster with closed loops instead of bare terminals? The network may actively form loops to contribute to the neuromast's overall structure and function, or else emerge as a consequence of the arbors navigating the intervening spaces between the hair cells and support cells. The preferential localization of the postsynaptic aggregates in the loops and an enrichment near the nodes of the arbor network suggest that the precise patterning of the network—with characteristic looping behaviors—is not merely a byproduct of neuromast architecture, but may contribute to the network's ability to gather sensory inputs through properly placed postsynaptic aggregates and to faithfully transmit sensory information to higher processing centers in the hindbrain (*Valera et al., 2021*). Further studies are necessary to elucidate the potential role of these topological features in regulating the distribution of synaptic bodies and optimizing robust neural transmission in the lateral-line microcircuit.

Recent studies have begun to dissect the molecular mechanisms of semaphorin signaling in axonal navigation. In explants derived from the murine olfactory bulb, Sema7A interacts directly with integrinβ1 to trigger downstream signaling effectors involving focal adhesion kinase and mitogen-activated protein kinases, which are critical regulators of the cytoskeletal network during axonal pathfinding (*Jeroen Pasterkamp et al., 2003*; *Myers and Gomez, 2011*). Moreover, the interaction between Sema7A and its receptor plexinC1 activates the rac1-cdc42-PAK pathway, which alters the actin

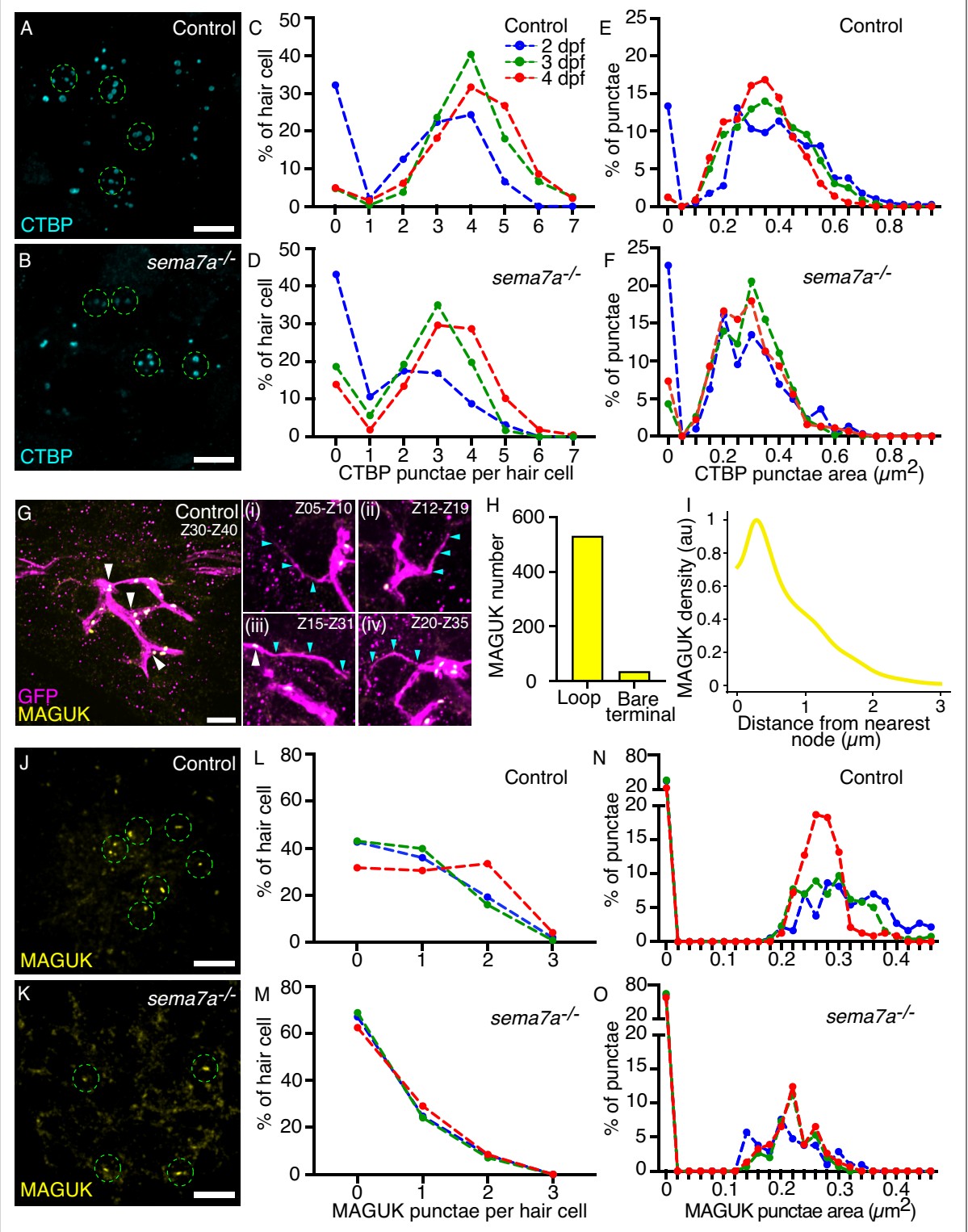

**Figure 6.** Sema7A deficiency impairs synaptic assembly. (**A, B**) Maximal-intensity projections of micrographs from a control and a *sema7a*-/- neuromast depict the presynaptic aggregates marked by CTBP (cyan). The approximate hair cell basal region is outlined by dashed green circles. (**C–F**) Plots quantitate the numbers of presynaptic aggregates (**C, D**) and their areas (**E, F**), in each hair cell across developmental stages. One hundred and fifty-two, 317, and 325 hair cells were analyzed from 2 dpf, 3 dpf, and 4 dpf control larvae, respectively. One hundred and sixty, 216, and 177 hair cells were analyzed from 2 dpf, 3 dpf, and 4 dpf *sema7a*-/- mutant larvae, respectively. (**G**) A micrograph generated from maximal-intensity projection of ten consecutive through-focus scans across Z-planes of a control 4 dpf neuromast depict the sensory arbors (magenta) and the postsynaptic aggregates

*Figure 6 continued on next page*

*Figure 6 continued*

marked by MAGUK (yellow, white arrowheads). The MAGUK punctae primarily occur on the loop and aggregate near the nodes. Panels (**i–iv**) depict maximal-intensity projection of consecutive through-focus scans of distinct Z-planes to show bare terminals. In each panel, the bare terminal (magenta, cyan arrowheads) rarely harbors postsynaptic aggregates (yellow, white arrowhead). (**H**) Quantification of the distribution of the MAGUK-punctae between the two topological features of the network depicts enrichment of postsynaptic densities on the loops over the bare terminals. (**I**) A histogram shows the distribution of MAGUK-punctae density against the distance from the node. The histogram is represented as a density trace graph (yellow). For both H and I, five hundred and sixty-seven MAGUK punctae were analyzed from 16 neuromasts of 4 dpf larvae. (**J, K**) Maximal-intensity projections from micrographs of a control and a *sema7a*⁻ᐟ⁻ neuromast depict the postsynaptic aggregates marked by MAGUK (yellow). The approximate hair cell basal region is outlined by dashed green circles. (**L–O**) Plots quantitate the numbers of postsynaptic aggregates (**L, M**) and their areas (**N, O**), in each hair cell across developmental stages. One hundred and fifty, 218, and 167 hair cells were analyzed from 2 dpf, 3 dpf, and 4 dpf control larvae, respectively. Ninety-seven, 141, and 141 hair cells were analyzed from 2 dpf, 3 dpf, and 4 dpf *sema7a*⁻ᐟ⁻ mutant larvae, respectively. Scale bars, 5 µm.

The online version of this article includes the following source data and figure supplement(s) for figure 6:

**Source data 1.** The source data represents the number and area of pre and post synaptic aggregates across genotypes.

**Figure supplement 1.** *sema7a*⁻ᐟ⁻ mutants display impaired synaptic assembly.

**Figure supplement 2.** Morphologies of presynaptic and postsynaptic assemblies and the distribution of postsynaptic aggregates across the sensory arbor network.

---

network to promote axonal guidance and synapse formation (*Inoue et al., 2018*; *O'Donnell et al., 2009*). We speculate that lateral-line sensory axons utilize similar mechanisms to sense and respond to Sema7A in the establishment of microcircuits with hair cells.

During early development of the posterior lateral line, the growth cones of sensory axons intimately associate with the prosensory primordium that deposits neuromasts along the lateral surface of the larva (*Gilmour et al., 2004*; *Metcalfe, 1985*). The growth of the sensory fibers with the primordium is regulated by diverse neurotrophic factors expressed in the primordium (*Gasanov et al., 2015*; *Schuster et al., 2010*). Impairing brain-derived or glial cell line-derived neurotrophic factors and their receptors perturbs directed axonal motility and innervation of hair cells (*Gasanov et al.,*

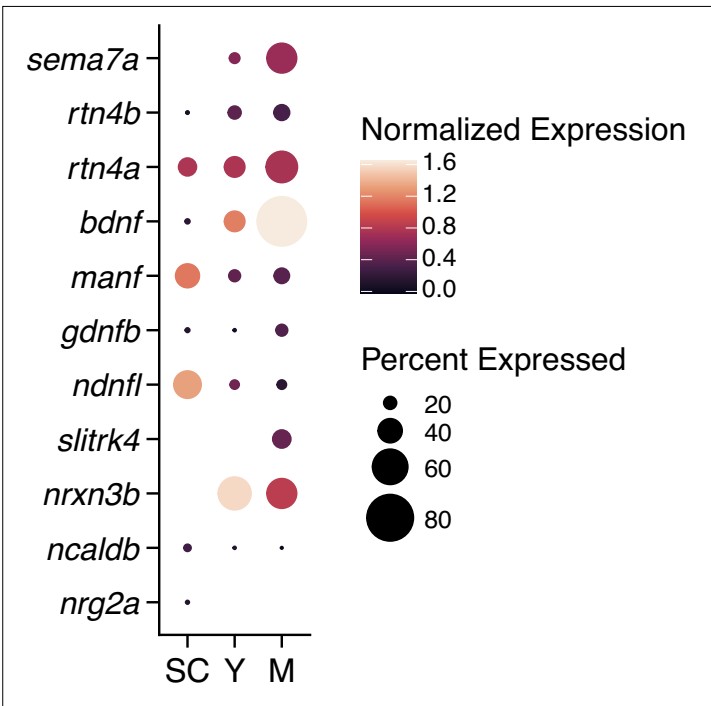

**Figure 7.** Expression of diverse neural guidance cues in developing neuromasts. Analysis of single cell RNA-sequencing data (*Lush et al., 2019*) shows the expression of multiple neural guidance genes that are expressed across young hair cells (Y), mature hair cells (M), and support cells (SC). Dot sizes and colors signify the proportion of cells in each cluster that express a gene and the average strength of expression(log[(counts/10,000)+1]), respectively.

*2015*; *Schuster et al., 2010*). Lateral-line hair cells are particularly enriched in brain-derived neurotrophic factor (*Germanà et al., 2010*) and other chemotropic cues (*Figure 7*). Because the sensory axons of *sema7a*[-/-] mutants reliably branch from the lateral-line nerve to reach hair cell clusters, these chemoattractive factors might compensate in part for the absence of Sema7A function. Although sensory axons are strict selectors of hair cell polarity and form synapses accordingly (*Nagiel et al., 2008*), diverse chemoattractant cues including Sema7A occur uniformly between hair cell polarities (*Germanà et al., 2010*). It seems likely that unbiased attractive signals bring sensory arbors toward hair cells, after which polarity-specific molecules on hair cell surfaces dictate the final association between axonal terminal and hair cells. Notch-delta signaling and its downstream effectors regulate hair cell polarity and the innervation pattern (*Jacobo et al., 2019*; *Dow et al., 2018*; *Ji et al., 2018*). It would be interesting to determine whether Notch-mediated polarity signatures on the hair cell's surface can refine the neuromast neural circuitry.

## Materials and methods

### Zebrafish strains and husbandry

Experiments were performed on 1.5–4 dpf zebrafish larvae in accordance with the standards of Rockefeller University's Institutional Animal Care and Use Committee (IACUC) with protocol number 22,081 H (PRV 19,076 H). Naturally spawned and fertilized eggs were collected, cleaned, staged, and maintained at 28.5 °C in system-water with 1 µg/mL methylene blue. Wild-type AB and heterozygous mutant *semaphorin7a*[sa24691] zebrafish were obtained from the Zebrafish International Resource Center. In addition, the following transgenic zebrafish lines were used: *Tg(myo6b:actb1-EGFP)* (*Kindt et al., 2012*), *Tg(neurod1:tdTomato)* (*Ji et al., 2018*), *Tg(cldnb:lyn-mScarlet)* (*Dalle Nogare et al., 2020*), and *TgBAC(neurod1:EGFP)* (*Obholzer et al., 2008*).

### Embryo dissociation and flow cytometry

Immediately before dissection, 4-day-old *Tg(myo6b:actb1-EGFP)* and wild-type larvae were anesthetized in 600 µM 3-aminobenzoic acid ethyl ester methanesulfonate in system water with 1 µg/mL methylene blue. The anterior half of each larva was removed with a fine surgical blade. We used approximately 400 GFP-positive and 50 GFP-negative larvae, the latter for controls of the gating procedure. Dissociation of the tails were performed using a protocol (*Baek et al., 2022*) with minor modification; detailed protocol is available upon request. The resuspended cells were incubated for 5 min on ice and sorted with an influx cell sorter (BD Biosciences, San Jose, CA, USA) with an 85 µm nozzle at 20 psi and 1 X DPBS (Sigma-Aldrich, St Louis, USA). Laser excitation was conducted at 561 nm and 488 nm. Distinct populations of GFP-positive hair cells were isolated based on forward scattering, lateral scattering, and the intensity of GFP fluorescence.

### RNA extraction and cDNA library preparation

Sorted cells were collected in a lysis-buffer solution supplemented with β-mercaptoethanol and RNA extraction was performed immediately using a standard protocol (RNeasy Plus Micro Kit; QIAGEN). The yield and quality of the product were measured with a bioanalyzer (Agilent 2100). Only samples with an RNA integrity score (RIN) greater than 8.0 were selected for the preparation of cDNA libraries using iScript cDNA Synthesis Kit (Bio-Rad). One nanogram of total RNA was amplified to several micrograms of cDNA.

### Reverse transcription-polymerase chain reactions (RT-PCR)

The *sema7a* gene expresses two transcript variants: variant 1, *sema7a-GPI* anchored, NM_001328508.1; and variant 2, *sema7a*[sec] secreted, NM_001114885.2. The variant 2 lacks several C-terminal exons, and its 3'-terminal exon extends past a splice site that is used in variant 1. The encoded variant 2 has a shorter and distinct C-terminus than variant 1. To individually identify these transcripts, each of them was PCR amplified from the cDNA library using the following primers: For variant 1, sema7aF = 5′-GGTTTTTCTGAGGCCATTCC-3′, sema7aR1=5′-GGCACTCGTGACAAATGCTA-3′; and for variant 2, sema7aF = 5′-GGTTTTTCTGAGGCCATTCC-3′, sema7aR2=5′-TGTGGAGAAAGTCACAAAGCA-3′. To detect *pvalb8* transcript we used the following primers: pvalb8F=5′-ATGTCTCTCACATCTATCCT-3′

and pvalb8R=5′-TCAGGACAGCACCATCGCCT-3′. To detect *s100t* transcript we used the following primers: s100tF = 5′-ATGCAGCTGATGATCCAGAC-3′ and s100tR = 5′-CTATTTCTTGTCATCCTTCT-3′.

## Immunofluorescence microscopy

For the immunofluorescence labeling of wholemounted wild-type control and *sema7a*[-/-] larvae, 1.5–4 dpf larvae were fixed overnight at 4 °C in 4% formaldehyde in phosphate-buffered saline solution (PBS) with 1% Tween-20 (1% PBST). Larvae were washed four times with 1% PBST for 15 min each, then placed for 2 hr in blocking solution containing PBS, 2% normal donkey serum (NDS), 0.5% Tween-20, and 1% BSA. Primary antibodies were diluted in fresh blocking solution and incubated with the specimens overnight at 4 °C. The primary antibodies were goat anti-Sema7A (1:200; AF1835, R&D Systems) (*Figure 1—figure supplement 4*; *Carcea et al., 2014*), rabbit anti-myosin VI (1:200; Proteus 25–6791), murine anti-GM130 (1:200; 610822, BD Transduction Laboratories), murine anti-CTBP (1:200; B-3, SC-55502, Santa Cruz Biotech), and rabbit anti-mCherry (1:200; GTX128508, GeneTex). Larvae were washed four times with 0.1% PBST for 15 min each. Alexa Fluor-conjugated secondary antibodies (Invitrogen, Molecular Probes) applied overnight at 1:200 dilutions in 0.2% PBST included anti-rabbit (488 nm), anti-goat (555 nm), and anti-mouse (647 nm). Larvae were then washed four times in 0.2% PBST for 15 min each and stored at 4 °C in VectaShield (Vector Laboratories).

For the immunofluorescence labeling of postsynaptic density in wholemounted wild-type control and *sema7a*[-/-] larvae, 2–4 dpf larvae were fixed with 4% formaldehyde in PBS for 4.5–6 hr at 4 °C. Larvae were washed five times with 0.01% PBST for 5 min each and then rinsed in distilled water for 5 min. The larvae were permeabilized with ice-cold acetone at –20 °C for 5 min. They were rinsed in distilled water for 5 min, washed five times with 0.01% PBST for 5 min each, and blocked overnight with PBS buffer containing 2% goat serum and 1% BSA. Murine anti-pan-MAGUK antibody (1:500; IgG1 NeuroMab, K28.86, 75–029 and MABN72, Millipore) was diluted in PBS containing 1% BSA, added to the larvae, and incubated for 3 hr at room temperature. Larvae were washed five times for 5 min each with 0.01% PBST. Alexa Fluor 555-conjugated anti-mouse secondary antibody (1:1000) diluted in PBS containing 1% BSA was added to the larvae and incubated for 2 hr at room temperature. Larvae were washed five times with 0.01% PBST for 5 min each, rinsed in distilled water for 5 min, and stored at 4 °C in VectaShield (Vector Laboratories).

Posterior segments of fixed larvae were mounted on a glass slide and imaged on an Olympus IX81 microscope with a microlens-based, super-resolution confocal microscope (VT-iSIM, VisiTech international) under 60 X and 100 X, silicone-oil objective lenses of numerical apertures 1.30 and 1.35, respectively. Static images at successive focal depths were captured at 200 nm intervals and deconvolved with the Microvolution software in ImageJ.

## Measurement of Sema7A and td-Tomato-positive sensory arbor intensities

The average Sema7A fluorescence intensities of the hair cells from 1.5 dpf, 2 dpf, 3 dpf, and 4 dpf neuromasts were measured by determining the mean gray level within each hair cell labeled by actin-GFP expression. The intensity distribution of Sema7A protein across the basal cell membrane was measured using the line profile tool in ImageJ (*Figure 1—figure supplement 2G–I*). The lines were drawn with 1 µm length, starting from the inside of the hair cell to the outside. For each hair cell membrane, the intensity values were scaled from 0 to 1 arbitrary units. The intensity distribution of Sema7A protein along the apicobasal axis of the hair cell was measured using the line profile tool in ImageJ (*Figure 1—figure supplement 2J*). Because hair cell lengths differed among samples, lines were drawn with 2 µm widths and varying lengths of 6–8 µm. For each hair cell, apicobasal length was scaled from 0 to 100 arbitrary units and the intensity values from 0 to 1 arbitrary units. The average Sema7A and td-Tomato-positive sensory arbor fluorescence intensities at the hair cell bases from 2 dpf, 3 dpf, and 4 dpf neuromasts were measured by determining the mean gray level within each hair cell basal area, denoted by the region underneath the hair cell nucleus (*Figure 1—figure supplement 3A–E*).

## Genotyping of mutant fish

The *semaphorin7a*[sa24691] allele harbors an A-to-T point mutation in the seventh exon of the *semaphorin7a* gene, which creates a premature stop codon (*Figure 2—figure supplement 1A*). Genomic

DNA was isolated from the tail fins of adult *semaphorin7a$^{sa24691}$* (*sema7a$^{sa91}$*) heterozygous mutant zebrafish. The mutant locus in the genome was PCR amplified using the following primers: sema7a$^{sa91}$F: 5′-AAAGCTGGAAAGCGAATCAA-3′ and sema7a$^{sa91}$R: 5′-ATATCCAAGGATCCGCCTCT-3′. The 734-base pair amplicons were sequenced, and heterozygous adults were propagated.

## Lipophilic styryl fluophore labeling of hair cells

Freely swimming 4 dpf control and *sema7a$^{-/-}$* larvae were exposed to the styryl fluorophore FM 4–64 (N-(3-triethylammoniumpropyl)–4-(6-(4-(diethylamino) phenyl) hexatrienyl) pyridinium dibromide, ThermoFischer Scientific) at 5 µM for 30 s, then rinsed three times in fresh embryo water and immediately mounted in low-melting-point agarose for live imaging. The average FM 4–64 intensities within the hair cells of 4 dpf neuromasts of both genotypes were measured by determining the mean gray level within each hair cell labeled by actin-GFP expression.

## Transient transgenesis with the hsp70l:sema7a$^{sec}$-mKate2 construct

The pDNR-LIB plasmid containing the *sema7a$^{sec}$* coding sequence was purchased from horizon (7140389, Perkin-Elmer). The coding sequence was PCR amplified using the following primers: sema7a$^{sec}$F: 5′-GGGGACAAGTTTGTACAAAAAAGCAGGCTTGATGATTCGACATTATTCT-3′ and sema7a$^{sec}$R: 5′-GGGGACCACTTTGTACAAGAAAGCTGGGTGC*TTT*GTGGAGAAAGTCACAAAGCA-3′. The italicized nucleotides denote the mutated stop codon. The amplicon was then cloned into the pDONR221 vector by BP recombination to generate the middle entry pME-sema7a$^{sec}$ vector. To generate the hsp70:sema7a$^{sec}$-mKate2 construct, gateway cloning was performed by combining the plasmids p5E-hsp70 (*Kwan et al., 2007*), pME-sema7a$^{sec}$, p3E-mKate2-myc no-pA (Addgene, 80812), and pDESTtol2pACrymCherry (Addgene, 64023) with LR Clonase II Plus (Invitrogen). Verified constructs (25 ng/µl plasmid DNA) were injected with Tol2 Transposase mRNA (approximately 25 ng/µl) into one-cell embryos. The transiently transgenic larvae—identified by the expression of red fluorescent protein (mCherry) in their lenses—were raised till 3 dpf, heat-shocked in a water bath at 37 °C for 1 hr, incubated at 28 °C for 1 hr to allow expression, and subsequently mounted for live imaging.

## Transient transgenesis with the hsp70l:sema7a-GPI-2A-mCherry construct

The *sema7a-GPI* coding sequence without the stop codon (*sema7a-GPI$^{no-stop}$*) was synthesized using gBlocks HiFi Gene Fragments (Integrated DNA Technologies). The *sema7a-GPI$^{no-stop}$* coding sequence was then cloned into the pDONR221 vector by BP recombination to generate the middle entry pME-sema7a-GPI$^{no-stop}$ vector, henceforth denoted as pME-sema7a-GPI. To generate the hsp70:sema7a-GPI-2A-mCherry construct, gateway cloning was performed by combining the plasmids p5E-hsp70 (*Kwan et al., 2007*), pME-sema7a-GPI, p3E-2A-mcherrypA (Addgene, 26031), and pDESTtol2pACryGFP (Addgene, 64022) with LR Clonase II Plus (Invitrogen). Verified constructs (25 ng/µl plasmid DNA) were injected with Tol2 Transposase mRNA (approximately 25 ng/µl) into one-cell embryos. The transiently transgenic larvae—identified by the expression of green fluorescent protein (GFP) in their lenses—were raised till 3 dpf, heat-shocked in a water bath at 37 °C for 1 hr, incubated at 28 °C for 1 hr to allow expression, and subsequently mounted for live imaging.

## Microscopy and volumetric rendering of living neuronal arbors and cells

Living larvae were dechorionated, anaesthetized in 600 µM 3-aminobenzoic acid ethyl ester methanesulfonate in system-water with 1 µg/mL methylene blue, and mounted in 1% low-melting-point agarose on a glass-bottom MetTek dish at specific stages. The larvae were maintained at 28 °C during imaging in a temperature-controlled stage top chamber (OKO Lab UNO-T). Static images at successive focal depths were captured at 200 nm intervals and deconvolved with Microvolution software in ImageJ (NIH). For time-lapse video microscopy, the ectopic *sema7a$^{sec}$* integration sites were imaged at intervals of 5 min as Z-stacks acquired with 1.0 µm steps with laser excitation at 488 nm and 561 nm under a 60 X, silicone-oil objective lens of numerical aperture 1.30. The three-dimensional datasets were processed and volume-rendered with the surface evolver and filament-tracer tools in Imaris (Bitplane, Belfast, UK). The Imaris workstation was provided by the Rockefeller University Bio-imaging Resource Center.

## Generation of pseudocolored trajectories of the dwell times of sensory neurites at the ectopic Sema7A^sec sources

Fluorescent intensity values of the sensory neurites and the ectopic Sema7A^sec sources were measured from the maximal-intensity projections at each time during the time-lapse video microscopy. The intensity values were normalized and binarized before summing across each timepoint for locations across the XY plane. The resulting pseudocolored trajectories represent the dwell times of the neurites and the Sema7A^sec cells through the 10 hr time-lapse video sequence.

## Generation of skeletonized networks

The labeled sensory axons were traced in three dimensions with ImageJ's semi-automated framework, SNT (*Arshadi et al., 2021*). Each trace depicts the skeletonized form of the posterior lateral-line nerve, the posterior lateral-line branch, and the point of arborization from which the sensory arbors radiate to contact the hair cells (*Figure 2D and E*; *Figure 2—figure supplement 2B*). The three-dimensional pixels or voxels obtained from the skeletonized sensory arbor traces were processed using custom code written in Python for visual representation and quantitative analysis.

## Generation of combined hair cell clusters and the corresponding combined skeletonized networks

The hair cell clusters of neuromasts from each developmental stage were aligned by the centers of their apices, which were identified by the hair bundles, and overlayed to generate combined hair cell clusters (*Figure 2—figure supplement 1E and F*) using custom Python code. The associated skeletonized sensory-arbor traces at each developmental stage were aligned similarly using custom Python code. The details of the procedures are available upon request and the corresponding code is available on GitHub.

## Quantification of sensory arbor distributions around hair cell clusters

The region from the center of the combined hair cell cluster to 60 µm in the post-cluster region was divided into 3600 concentric sections of equal area. Each section, as depicted with a distinct color shade, had an area of $\pi$ µm$^2$. Sensory arbor density was defined as $\mathrm{Log}_{10}$[(Area occupied by the arbors in each section)/ (Area of each section)]. The arbor density was then plotted as a histogram against distance from the center of the combined hair cell cluster. Each bin of the histogram represents an area of $\pi$ µm$^2$ (*Figure 2—figure supplement 2A*). The histograms were finally represented as density trace graphs with Kernel Density Estimation (KDE) in Python.

## Quantification of contact between sensory arbors and hair cell clusters

For each hair cell cluster, the voxels of the sensory arbor traces that were inside or within 0.5 µm—the average neurite radius—of the cluster boundary were denoted as arbors in contact with their corresponding hair cell cluster (*Figure 2—figure supplement 2B*). The percentage of arbors in contact with each hair cell cluster was defined as: (Number of voxels that remained inside the cluster boundary/ Total number of voxels in the arbor) X 100. The data were analyzed through an automated pipeline with custom Python code.

## Quantification of the sensory arborization network topology

We developed a custom Python script (snt_topology.py; available on GitHub) to convert the skeletonized sensory arborization networks to network graphs (NetworkX, v3.1). We first used a custom ImageJ macro to convert SNT skeletons to csv files, then generated the network graph from the individual arbor paths. As noted previously, we defined five topological attributes: (1) nodes—junctions where arbors branch, converge, or cross; (2) loops—minimal arbor cycles forming topological holes; (3) bare terminals—arbors with free ends; (4) node degree—the number of first-neighbor nodes incident to a node; and (5) arbor curvature—the total curvature of a bare terminal (*Figure 3A*). To enumerate loops and quantify their lengths and node numbers, we computed the minimum cycle basis for each network graph. The relative organ height of each node in a graph was defined in 3D using *Equation 1*:

$$\left( \vec{\mathrm{u}} \cdot \vec{\mathrm{v}} \right) / \left\| \vec{\mathrm{v}} \right\|^2, \tag{1}$$

where $\vec{u}$ is the vector from the arborization initiation point to a node and $\vec{v}$ is the vector from the arborization initiation point to the hair cell cluster apex. The relative occurrence of nodes (*Figure 3— figure supplement 1A–C*) was normalized by both the number of hair cells per neuromast and the total number of neuromasts, for comparison across ages and genotypes. The relative number of bare terminals and loops (*Figure 3B*; *Figure 3—figure supplement 1D, E*) was normalized by the number of hair cells per neuromast and scaled between 0 and 1. The density was then normalized by the total number of neuromasts imaged for each developmental stage and genotype. For *Figure 3C* and *Figure 3—figure supplement 1F, G*, the normalized occurrence was computed as the number of arbors of a given length that formed either a bare terminal or a loop, divided by the total number of arbors of either feature of that length. Lastly, we defined bare terminals as having either high or low curvature (*Figure 3F*; *Figure 3—figure supplement 2E and F*) if their absolute total signed curvature [*Equation 2*] in three dimensions was either higher or lower than $\pi/6$, respectively:

$$\left| \int \frac{dT}{ds} \cdot N(s)\, ds \right|, \tag{2}$$

where $s$ is the arc-length parameter of the bare terminal, $T(s)$ is the unit tangent vector at each point along the arbor, $N(s)$ is the unit normal vector at each point, and the integral is evaluated from the terminal's branching node to the arbor's tip. For a closed curve forming a full loop, this value will equal $2\pi$. In practice, we first fit B-splines (scipy.interpolate.splprep; SciPy, v1.10.1) in 3D to the skeletons for each bare terminal before evaluating their derivatives (scipy.interpolate.splev) and integrating (scipy.integrate.simpson) over their lengths.

### Quantification of aberrant sensory arborization in ectopic Sema7A^sec expression

The sensory arbor networks were traced in three dimensions with SNT. The surface of each of the ectopically expressing Sema7A^sec-mKate2 cells was represented by a single point that was closest to the terminal point of the guided neurite. Distances between two points were calculated using standard mathematical operations.

### Quantification of presynaptic and postsynaptic assembly number and area

Maximal-intensity projections of consecutive through-focus scans encompassing the entire neuromast of control and *sema7a^-/-* mutant larvae at 2 dpf, 3 dfp, and 4 dpf were utilized to capture the presynaptic and postsynaptic assemblies. The number of the synaptic densities across genotypes were measured using ImageJ's cell counter framework. Area of the synaptic densities from the maximal-intensity projections were measured using ImageJ's polygon selection tool (*Figure 6—figure supplement 2A–D*).

### Quantification of the distribution of postsynaptic aggregates along the loops and the bare terminals of the sensory arbor networks

Four-day-old *TgBAC(neurod1:EGFP)* larvae were fixed and stained for sensory arbors and postsynaptic aggregates marked by GFP and MAGUK, respectively. The GFP-positive sensory arbors were traced in three dimensions with ImageJ's semi-automated framework, SNT. The topological attributes of the arbor network, such as the loop and the bare terminal, were determined as described previously. The localizations of the MAGUK punctae along the three-dimensional arbor network were measured using ImageJ's cell counter framework (*Figure 6—figure supplement 2E and F*). We measured 567 MAGUK punctae from 16 sensory arbor networks. Whether the MAGUK-punctae were associated with a loop or a bare terminal, we categorized them in two groups and quantified their distribution across 16 neuromasts.

### Quantification of the distribution of postsynaptic-aggregate density along the distance from the nearest node

The three-dimensional space surrounding each node of the arbor network were divided into concentric shells of equal volume (*Figure 6—figure supplement 2E, F*). For each node, we computationally

generated 100,000 concentric shells that reached a 10 μm radius. We measured the number of MAGUK punctae located within each of those concentric shells from their nearest nodes.

## Statistical analysis

Data visualization and statistical analysis were conducted with GraphPad Prism (Version 9) and Python (v3.10). The Mann-Whitney test was used for hypothesis testing unless otherwise noted, and the statistical details are given in the respective figure legends.

## Acknowledgements

The authors thank Nicolas Velez and Emily Atlas for valuable discussions. Samantha Campbell provided expert fish husbandry. Katie Kindt kindly provided various strains of zebrafish. Image processing benefitted from facilities at the Bio-Imaging Resource Center. AD was supported by a Kavli Neural Systems Institute Postdoctoral Fellowship from The Rockefeller University. CCR is supported by National Science Foundation Graduate Research Fellowship Grant No. 1946429. AJ, SPP, and LMS were supported by Howard Hughes Medical Institute, of which AJH is an Investigator.

## Additional information

### Funding

| Funder | Grant reference number | Author |
|---|---|---|
| Rockefeller University | Kavli Neural Systems Institute Postdoctoral Fellowship | Agnik Dasgupta |
| Howard Hughes Medical Institute | | AJ Hudspeth |
| National Science Foundation Graduate Research | 1946429 | Caleb C Reagor |

The funders had no role in study design, data collection and interpretation, or the decision to submit the work for publication.

### Author contributions

Agnik Dasgupta, Conceptualization, Data curation, Formal analysis, Supervision, Validation, Investigation, Visualization, Methodology, Writing – original draft, Project administration, Writing – review and editing; Caleb C Reagor, Software, Formal analysis, Visualization, Methodology, Writing – review and editing; Sang Peter Paik, Software, Formal analysis, Visualization, Methodology; Lauren M Snow, Conducted preliminary experiments to validate the Semaphorin7A antibody; Adrian Jacobo, Supervision, Visualization, Methodology, Performed Semaphorin7A intensity measurements; AJ Hudspeth, Resources, Supervision, Funding acquisition, Project administration, Writing – review and editing

### Author ORCIDs

Agnik Dasgupta (ID) https://orcid.org/0000-0003-0860-1006
Caleb C Reagor (ID) https://orcid.org/0000-0002-8304-1267
AJ Hudspeth (ID) https://orcid.org/0000-0002-0295-1323

### Ethics

Experiments were performed on 1.5 to 4 dpf zebrafish larvae in accordance with the standards of Rockefeller University's Institutional Animal Care and Use Committee (IACUC) with protocol number 22081-H (PRV 19076-H).

Reviewer #1 (Public Review): https://doi.org/10.7554/eLife.89926.4.sa1
Reviewer #2 (Public Review): https://doi.org/10.7554/eLife.89926.4.sa2
Reviewer #3 (Public Review): https://doi.org/10.7554/eLife.89926.4.sa3

Reviewer #4 (Public Review): https://doi.org/10.7554/eLife.89926.4.sa4
Author response https://doi.org/10.7554/eLife.89926.4.sa5

# Additional files

## Supplementary files
MDAR checklist

## Data availability
All data generated or analysed during this study are included in the manuscript and supporting files; source data files have been provided for relevant figures. The processed version of the imaging data have been included in the manuscript. Raw imaging data are available for this paper and will be shared by the corresponding author upon reasonable request. The codes generated during this study are available on GitHub (Source code 1: https://github.com/agnikdasgupta/Sema7A_regulates_neural_circuitry; copy archived at *Reagor, 2023*).

The following previously published datasets were used:

| Author(s) | Year | Dataset title | Dataset URL | Database and Identifier |
|---|---|---|---|---|
| Lush ME, Diaz DC, Koenecke N, Baek S, Boldt H, St Peter MK, Gaitan-Escudero T, Romero-Carvajal A, Busch-Nentwich EM, Perera AG, Hall KE, Peak A, Haug JS, Piotrowski T | 2019 | Zebrafish Neuromast scRNA-seq | https://piotrowskilab.shinyapps.io/neuromast_homeostasis_scrnaseq_2018/ | Piotrowski Lab Shiny Apps, neuromast_homeostasis_scrnaseq_2018 |
| Baek S, Tran NTT, Diaz DC, Tsai YY, Navajas Acedo J, Lush ME, Piotrowski T | 2022 | Neuromast Regeneration scRNA-seq | https://piotrowskilab.shinyapps.io/neuromast_regeneration_scRNAseq_pub_2021/ | Piotrowski Lab Shiny Apps, neuromast_regeneration_scRNAseq_pub_2021 |

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
