## [Editor Report · eLife assessment]

The **valuable** findings by Dasgupta et al demonstrate the role of Sema7a in fine tuning the morphology of the microcircuit between afferent axons and sensory hair cells in the lateral line organ. The loss and gain of function evidence provides **solid** support for a role for Sema7a in this process. Additional work is needed to determine the role for different isoforms in Sema7a-mediated synapse formation and chemoattraction as well as cell type specificity.

---

## [Referee Report · Reviewer #1 (Public Review)]

Dasguta et al. have dissected the role of Sema7a in fine tuning of a sensory microcircuit in the posterior lateral line organ of zebrafish. They attempt to also outline the different roles of a secreted verses membrane-bound form of Sema7a in this process. Using genetic perturbations and axonal network analysis, the authors show that loss of both Sema7a isoforms causes abnormal axon terminal structure with more bare terminals and fewer loops in contact with presynaptic sensory hair cells. Further, they show that loss of Sema7a causes decreased number and size of both the pre- and post-synapse. Finally, they show that overexpression of the secreted form of Sema7a specifically can elicit axon terminal outgrowth to an ectopic Sema7a expressing cell. Together, the analysis of Sema7a loss of function and overexpression on axon arbor structure is fairly thorough and revealed a novel role for Sema7a in axon terminal structure.

---

## [Referee Report · Reviewer #2 (Public Review)]

In this work, Dasgupta et al. investigate the role of Sema7a in the formation of peripheral sensory circuit in the lateral line system of zebrafish. They show that Sema7a protein is present during neuromast maturation and localized, in part, to the base of hair cells (HCs). This would be consistent with pre-synaptic Sema7a mediating formation and/or stabilization of the synapse. They use sema7a loss-of-function strain to show that lateral line sensory terminals display abnormal arborization. They provide highly quantitative analysis of the lateral line terminal arborization to show that a number of specific topological parameters are affected in mutants. Next, they ectopically express a secreted form of Sema7a to show that lateral line terminals can be ectopically attracted to the source. Finally, they also demonstrate that the synaptic assembly is impaired in the sema7a mutant. Overall, the data are of high quality and properly controlled. The availability of Sema7a antibody is a big plus, as it allows to address the endogenous protein localization as well to show the signal absence in the sema7a mutant. The quantification of the arbor topology should be useful to people in the field who are looking at the lateral line as well as other axonal terminals.

---

## [Referee Report · Reviewer #3 (Public Review)]

The data reported here demonstrate that Sema7a defines the local behavior of growing axons in the developing zebrafish lateral line. The analysis is sophisticated and convincingly demonstrates effects on axon growth and synapse architecture. Collectively, the findings point to the idea that the diffusible form of sema7a may influence how axons grow within the neuromast and that the GPI-linked form of sema7a may subsequently impact how synapses form, though additional work is needed to strongly link each form to its' proposed effect on circuit assembly.

Comments on latest version:

The authors comprehensively and appropriately addressed most of the reviewers' concerns. In particular, they added evidence that hair cells express both Sema7A isoforms, showed that membrane bound Sema7A does not have long range effects on guidance, demonstrated how axons behave close to ectopic Sema7A, and analyzed other features of the hair cells that revealed no strong phenotypes. The authors also softened the language in many, but not all places. Overall, I am satisfied with the study as a whole.

---

## [Referee Report · Reviewer #4 (Public Review)]

This study provides direct evidence showing that Sema7a plays a role in the axon growth during the formation of peripheral sensory circuits in the lateral-line system of zebrafish. This is a valuable finding because the molecules for axon growth in hair-cell sensory systems are not well understood. The majority of the experimental evidence is convincing, and the analysis is rigorous. The evidence supporting Sema7a's juxtracrine vs. secreted role and involvement in synapse formation in hair cells is less conclusive. The study will be of interest to cell, molecular and developmental biologists, and sensory neuroscientists.

---

## [Author Response]

The following is the authors’ response to the previous reviews.

**eLife assessment**
Dasgupta and colleagues make a valuable contribution to the understanding how the guidance factor Sema7a promotes connections between mechanosensory hair cells and afferent neurons of the zebrafish lateral line system. The authors provide solid evidence that loss of Sema7a function results in fewer contacts between hair cells and afferents through comprehensive quantitative analysis. Additional work is needed to distinguish the effects of different isoforms of Sema7a to determine whether there are specific roles of secreted and membrane bound forms.
**Public Reviews:**

**Reviewer #1 (Public Review):**
Dasguta et al. have dissected the role of Sema7a in fine tuning of a sensory microcircuit in the posterior lateral line organ of zebrafish. They attempt to also outline the different roles of a secreted verses membrane-bound form of Sema7a in this process. Using genetic perturbations and axonal network analysis, the authors show that loss of both Sema7a isoforms causes abnormal axon terminal structure with more bare terminals and fewer loops in contact with presynaptic sensory hair cells. Further, they show that loss of Sema7a causes decreased number and size of both the pre- and post-synapse. Finally, they show that overexpression of the secreted form of Sema7a specifically can elicit axon terminal outgrowth to an ectopic Sema7a expressing cell. Together, the analysis of Sema7a loss of function and overexpression on axon arbor structure is fairly thorough and revealed a novel role for Sema7a in axon terminal structure. However, the connection between different isoforms of Sema7a and the axon arborization needs to be substantiated. Furthermore, the effect of loss of Sema7a on the presynaptic cell is not ruled out as a contributing factor to the synaptic and axon structure phenotypes. These issues weaken the claims made by the authors including the statement that they have identified dual roles for the GPI-anchored verses secreted forms of Sema7a on synapse formation and as a chemoattractant for axon arborization respectively.
**Reviewer #2 (Public Review):**
In this work, Dasgupta et al. investigates the role of Sema7a in the formation of peripheral sensory circuit in the lateral line system of zebrafish. They show that Sema7a protein is present during neuromast maturation and localized, in part, to the base of hair cells (HCs). This would be consistent with pre-synaptic Sema7a mediating formation and/or stabilization of the synapse. They use sema7a loss-of-function strain to show that lateral line sensory terminals display abnormal arborization. They provide highly quantitative analysis of the lateral line terminal arborization to show that a number of specific topological parameters are affected in mutants. Next, they ectopically express a secreted form of Sema7a to show that lateral line terminals can be ectopically attracted to the source. Finally, they also demonstrate that the synaptic assembly is impaired in the sema7a mutant. Overall, the data are of high quality and properly controlled. The availability of Sema7a antibody is a big plus, as it allows to address the endogenous protein localization as well to show the signal absence in the sema7a mutant. The quantification of the arbor topology should be useful to people in the field who are looking at the lateral line as well as other axonal terminals. I think some results are overinterpreted though. The authors state: "Our findings demonstrate that Sema7A functions both as a juxtracrine and as a secreted cue to pattern neural circuitry during sensory organ development." However, they have not actually demonstrated which isoform functions in HCs (also see comments below). In addition, they have to be careful in interpreting their topology analysis, as they cannot separate individual axons. Thus, such analysis can generate artifacts. They can perform additional experiments to address these issues or adjust their interpretations.
**Reviewer #3 (Public Review):**
The data reported here demonstrate that Sema7a defines the local behavior of growing axons in the developing zebrafish lateral line. The analysis is sophisticated and convincingly demonstrates effects on axon growth and synapse architecture. Collectively, the findings point to the idea that the diffusible form of sema7a may influence how axons grow within the neuromast and that the GPI-linked form of sema7a may subsequently impact how synapses form, though additional work is needed to strongly link each form to its' proposed effect on circuit assembly.The revised manuscript is significantly improved. The authors comprehensively and appropriately addressed most of the reviewers' concerns. In particular, they added evidence that hair cells express both Sema7A isoforms, showed that membrane bound Sema7A does not have long range effects on guidance, demonstrated how axons behave close to ectopic Sema7A, and analyzed other features of the hair cells that revealed no strong phenotypes. The authors also softened the language in many, but not all places. Overall, I am satisfied with the study as a whole.
**Reviewer #4 (Public Review):**
This study provides direct evidence showing that Sema7a plays a role in the axon growth during the formation of peripheral sensory circuits in the lateral-line system of zebrafish. This is a valuable finding because the molecules for axon growth in hair-cell sensory systems are not well understood. The majority of the experimental evidence is convincing, and the analysis is rigorous. The evidence supporting Sema7a's juxtracrine vs. secreted role and involvement in synapse formation in hair cells is less conclusive. The study will be of interest to cell, molecular and developmental biologists, and sensory neuroscientists.
**Recommendations for the authors:**

**Reviewer #1 (Recommendations For The Authors):**
In their revised manuscript, Dasgupta et al. have provided further experiments to address the role of Sema7a (sec and GPI-anchored) in regulating axon guidance in the lateral line system. Specifically, the inclusion of the heat shock controls and FM labeling to show hair cell mechanotransduction were crucial to interpretation of the results. However, there are still concerns about the specificity of the results. My primary concern is if the change in axon patterning is specifically due to loss of Sema7a in the mutant hair cells. These animals are morphologically very abnormal and, in the rebuttal, the authors state that hair cell number is reduced. This is not quantified in the manuscript and should be included.

Thank you for this suggestion. We have included the data in the manuscript in lines 137-139, in Figure 2—figure supplement 1B, and in the source data for Figure 2 and Figure 2-figure supplements.

If there is not a function for Sema7a in hair cells themselves, why is the number reduced?

The sema7a-/- homozygous mutants are not viable and they die by 6 dpf. The loss of Sema7A protein produce other developmental defects including brain edema and a curved body axis. We believe a slight but not significant decrease in hair cell number may arise from a minute developmental delay in the morphogenesis of the neuromast. We have accordingly quantified our data at three distinct developmental stages-at 2 dpf, 3 dpf, and 4 dpf-and have incorporated them in the revised manuscript.

Additionally, FM data should be quantified and presented in animals without a transgene in the same excitation/emission spectra for clearer interpretation of the staining.

We have quantified the intensities of labeling with FM 4-64 styryl dye from the control and the sema7a-/- mutant larvae and incorporated the data in lines 139-146, in Figure 2—figure supplement 1D, and in source data for Figure 2 and Figure 2-figure supplements. We Kept the transgenes to concurrently show the arborization phenotype, hair cell morphology, and the FM 4-64 incorporation between the genotypes.

Rescue analysis using the myo6d promotor would allow the authors to ensure that the axon deficits can be rescued by putting Sema7a back into the sensory hair cells. Transient transgenesis could be useful for this approach and would not require the creation of a stable line. This could be done with both forms of Sema7a allowing the true assessment of whether or not the secreted and GPI-anchored form have disparate functions as claimed in lines 418424.

Although we recognize the importance of the rescue of the sema7a-/- mutant phenotype with the sema7asec and the sema7aGPI transcripts, it is not possible for us to perform that experiment at the moment, for the first author will leave the lab next week. However, he plans to continue work on this project as an independent investigator to dissect the individual roles of the transcript variants in specifying the pattern of sensory arborization, a project that includes generation of transcript-specific knockout animals and rescue experiments with stable transgenic fish lines.

Other concerns:(1) The timeline of the heat shock experiment is confusing to me and, therefore, it makes me question the specificity of those results. Based on the speed of axon outgrowth and the time necessary for transcription and translation after heat shock induction of the transgene, it is unclear to me how the axon growth defects could occur in the timeline provided. Imaging two hours after the start of the heat shock is very rapid and speaks to either an indirect effect of the transgenesis on the axon growth or a leaky promotor/induction paradigm. It is possible I am just misunderstanding the set up but, from what I could gather, the imaging is being done 2 hrs after the start of the heat shock. This should be clarified.

The axons of the zebrafish posterior lateral line migrate relatively fast. The pioneering axons migrate at around 120 μm/hour (Sato et. al., 2010) and the follower axons migrate at almost 30-80 μm/hour (Sato et. al., 2010). The heat-shock promoter that we have utilized, hsp70l, is highly effective in inducing gene expression and subsequent protein formation within 30 to 60 mins. We believe an hour of heat shock and an hour of incubation post heat shock is sufficient to induce directed axon migration to a distance that spans from 27 μm to 140 μm.

We strongly believe that the directed arborization of the sensory axons towards the Sema7Asec source is not due to an indirect effect of transgenesis or leaky promoter induction, as in all 18 of the injected but not heat-shocked control larvae we did not observe ectopic Sema7Asec expression, and no aberrant projection was formed from the sensory arbor network. We highlight this observation in lines 297-299 and in Figure 4E.

Sato et. al., 2010: Single-cell analysis of somatotopic map formation in the zebrafish lateral line system. Developmental Dynamics 239:2058–2065, 2010.

Similarly, it would help to clarify if t(0) in the figure is the onset of the heat shock or onset of imaging two hours after the heat shock is started.

The t=0 hour in the Figure 4I denotes the onset of imaging two hours after the heat shock began. We have clarified this in the manuscript in lines 1155-1156.

(2) In the rebuttal, the line numbers cited do not match up with the appropriate text, I believe.

We have corrected this and updated the manuscript.

(3) Some of the supplemental figures are not mentioned in the text, or I could not find them. For example: Figure 1 supplement 2J.

Thank you for pointing this. We have corrected the manuscript, and the new information is added in line 114.

(4) Table 1 statistics: were these adjusted for multiple comparisons using a bonferroni correction or something similar? This is necessary for statistical significance to be meaningful.

We did not adjust the p-values for multiple comparisons because the values correspond to only three or four statistical tests per experiment, strongly indicating the unlikelihood of erroneous significance due solely to multiple tests.

(5) Figure 1I and 1-S3 - The legend states a positive correlation between axonal signal and sema7A signal. Correlations are 0.5, 0.6, and 0.4 (2,3, 4dpf). This is not a convincing positive correlation. At best this is no to a very weak positive correlation.

In lines 122-126 we mention that the basal association of the sensory arbors shows a positive correlation with Sema7A accumulation. We never emphasize on the strength of the correlation. However, a consistent positive correlation at three different developmental stages suggests that progressive Sema7A accumulation at the base of the hair cells may guide the sensory arbors to increasingly associate themselves with the hair cells.

**Reviewer #2 (Recommendations For The Authors):**
I am a bit disappointed that the authors elected not to experimentally address the issue raised by all reviewers: whether the secreted or membrane bound isoform is active in hair cells. They rather decided to change their interpretation in the text. It is fine, given the eLife review structure. However, that would make the manuscript much stronger. Other issues were adequately addressed through textual changes as well.

Although we recognize the importance of the rescue of the sema7a-/- mutant phenotype with the sema7asec and the sema7aGPI transcripts, it is not possible for us to perform that experiment at the moment, for the first author will leave the lab next week. However, he plans to continue work on this project as an independent investigator to dissect the individual roles of the transcript variants in specifying the pattern of sensory arborization, a project that includes generation of transcript-specific knockout animals and rescue experiments with stable transgenic fish lines.

**Reviewer #3 (Recommendations For The Authors):**
Overall, I am satisfied with the study as a whole and just have a few minor comments that remain to be addressed.(1) Although the authors say that they added appropriate no plasmid/heatshock-only and plasmid-only/no heatshock controls, these results need to be presented more clearly, as they are separated in the paper and only one was quantified (i.e. 100% of embryos showed no defect). Please just make it clear that no defects were observed in either control for either experiment (both secreted and membrane bound ectopic expression).

We have clearly stated this information in lines 297-299 and 343-345.

(2) Please add a compass to Fig. 1A to indicate the orientation of the neuromast. It would also be helpful to add labels for developmental ages to all of the figures, rather than making the reader look it up in the legend.

We have updated the Figure 1A and the corresponding figure legend in lines 882883 . We have denoted the larval age in the figure legends to keep the individual images uncluttered.

(3) For the RT-PCR experiments in Figure 1, no negative control was included to show that supporting cell or neuronal genes are not detected in the purified hair cells and v.v. that neither isoform is detected in supporting cells or neurons. I ask only because there is a lot of immune-signal outside of the hair cells and I am curious whether that is secreted or might come from other cell types. For neurons and supporting cells, simply demonstrating absence of Sema7a overall would suffice.

We have utilized the transgenic line Tg(myo6b:actb1-EGFP) that expresses the fluorophore GFP specifically in the hair cells of the neuromast. Unfortunately, we do not possess a transgenic line that reliably and specifically labels the support cells in the neuromast. Hence, in our sorting experiment the GFP-negative cells that are collected from the trunk segments of the larvae contain all the non-hair cells including epidermal cells, neuronal cells, and immune cells etc. Such a mixture of varied cellular identity may not serve as a reliable negative control.

In Figure 7, we have plotted the normalized expression values of the sema7a gene in the neuromast. The plot clearly depicts that the source of Sema7A is the young and the mature hair cells, not the support cells. We further confirm this observation by

immunostaining where the Sema7A signal is highly restricted to the hair cells and not in any other cell in the neuromast (Figure 1E). Immunostaining further demonstrates that the lateral line sensory arbors also do not produce the Sema7A protein (Figure 1H; Video 1).

We agree with the reviewer that there are diverse immune cells, including macrophages in and around the neuromast. These macrophages are dynamic and possess highly ramified structure (Denans et. al., 2022). In all our Sema7A immunostainings, we never observed structures that resemble macrophages. Albeit we cannot confirm that Sema7A is not expressed in a distant immune cell, but we highly doubt that signal coming from immune cells is impacting hair cell innervation by the sensory arbors during homeostatic development.

Denans et. al., 2022: Nature Communications volume 13, Article number: 5356 (2022).

(4) In Figure 1, Supplement 4, I do not see the immunogen labeled in blue.

We have corrected the figure legend. The immunogenic region of the Sema7A protein is now clearly denoted in the figure legend of Figure 1—figure supplement 4.

(5) In Figure 2, please add a control image as requested, as that enables direct comparison. There is ample room in the figure.

We have updated the Figure 2 and made the suggested change.

(6) In Figure 2, Supplement 1, the FM4-64 data are not presented in a quantified fashion. Please report at least how many embryos showed reliable uptake and preferably how many hair cells per embryo showed reliable uptake.

We have quantified the FM 4-64 intensities in control and sema7a-/- mutant larvae. The new data is added to the manuscript in lines 142-146, 577-579 , and in Figure 2—figure supplement 1D.

(7) In Figure 3, there seems to be a typo in the figure legend: "mutants in the same larvae" does not make sense to me.

We have corrected the error. The modified statement is represented in lines 10671068.

(8) The text should refer more explicitly to the statistical tests reported in Table 1, i.e. as the results are presented.

In lines 1105 and 1109, we clearly state the statistical tests that were performed.

(9) In Figure 6, Supplement 1, please show the raw data points not just the bar graphs

We have updated the Figure 6—figure supplement 1.

(10) Minor point: the authors state that they addressed the distance over which secreted Sema7A may act, but this was not evident to me in the text. Please make this finding clearer.

We have clarified this information in lines 310-311.

(11) Finally, the discussion contains a statement that is not supported by the data: "We have discovered dual modes of Sema7A function in vivo." They have discovered evidence that there are two isoforms, that loss of both disrupts connectivity, and that overexpression of only the secreted form can elicit growth from a distance. However, there is no direct evidence that the membrane-bound form is responsible for local effects. It is formally possible still that the phenotypes are a result of dual roles for the secreted form. It is clear that another manuscript is forthcoming that will expand on the role of the transmembrane form, but for this manuscript, the authors should make firm conclusions only about the data presented herein.

Thank you for this suggestion. We have modified the manuscript in lines 425-434.

**Reviewer #4 (Recommendations For The Authors):**
The authors have made significant changes to the manuscript based on the comments of the reviewers. It is now suitable for publication.